# Decipher the complexity of cis-regulatory regions by a modified Cas9

**Steven Kirchner**  *, **Stefanie Reuter, Anika Westphal, Ralf Mrowka**

Experimental Nephrology Group, KIM III, Universitätsklinikum Jena, Jena, Germany

* steven.kirchner@uni-jena.de

## Abstract

### Background

Understanding complex mechanisms of human transcriptional regulation remains a major challenge. Classical reporter studies already enabled the discovery of *cis*-regulatory elements within the non-coding DNA; however, the influence of genomic context and potential interactions are still largely unknown. Using a modified Cas9 activation complex we explore the complexity of *renin* transcription in its native genomic context.

### Methods

With the help of genomic editing, we stably tagged the native *renin* on chromosome 1 with the *firefly luciferase* and stably integrated a programmable modified Cas9 based *trans*-activation complex (SAM-complex) by lentiviral transduction into human cells. By delivering five specific guide-RNA homologous to specific promoter regions of *renin* we were able to guide this SAM-complex to these regions of interest. We measured gene expression and generated and compared computational models.

### Results

SAM complexes induced activation of *renin* in our cells after *renin* specific guide-RNA had been provided. All possible combinations of the five guides were subjected to model analysis in linear models. Quantifying the prediction error and the calculation of an estimator of the relative quality of the statistical models for our given set of data revealed that a model incorporating interactions in the proximal promoter is the superior model for explanation of the data.

### Conclusion

By applying our combined experimental and modelling approach we can show that interactions occur within the selected sequences of the proximal *renin* promoter region. This combined approach might potentially be useful to investigate other genomic regions. Our findings may help to better understand the transcriptional regulation of human *renin*.

**Data Availability Statement:** All relevant data are within the paper and its Supporting Information files.

**Funding:** Grant Support: Funding: Bundesministerium für Bildung und Forschung

(BMBF) grant: FKZ 400 01EK1612B to RM. The funders had no role in study design, data collection and analysis, decision to publish, or preparation of the manuscript.

**Competing interests:** The authors have declared that no competing interests exist.

## Introduction

Transcriptional regulation of genes is one of the key points for gene expression in general. *Cis*-regulatory elements are regions of non-coding DNA that regulate the transcription of neighbouring genes and, among others, serve as a binding site for *trans*-factors. These elements form complex systems. By regulating these complex systems of thousands of genes the morphological maturation of cells as well as their differentiated function are made possible [1]. The understanding of these genomic *cis*-regulatory networks is therefore of great significance. There are examples of mutations of *cis*-regulatory promoter regions leading to severe diseases [2]. An example of a gene and its regulation that has been focused on by numerous research initiatives for many years is the human *renin* (*REN*). *REN* is considered to be a key enzyme in the renin-angiotensin-aldosterone system (RAAS). RAAS is a vital system of the human body, as it maintains plasma sodium concentration, arterial blood pressure and extracellular volume [3]. Abnormal activation of the RAAS can contribute to the development of hypertension, cardiac hypertrophy, and heart failure [4, 5]. According to the WHO about 1.13 billion people worldwide have hypertension which is one of the major causes of premature death [6]. Hence, the understanding of transcriptional regulation is potentially important to understand basic principles of many cardio-vascular diseases. Although the main effector molecule of RAAS is angiotensin II (ANG II), the regulation of ANG II and its precursors is mainly regulated by the expression of *REN* [7].

Important *cis*-regulatory elements could already be identified for *REN*. Most of the information about those elements could be obtained by cell culture experiments using classical reporter assays or experiments with transgenic mice [8, 9]. To perform classical reporter assays, restriction pieces of the DNA region under investigation are cloned in vectors. The sequences are followed by downstream reporter genes such as *green fluorescent protein* (GFP) or *Luciferase*, whose expression levels can be quantified in different ways. After transfection of the vectors in cells, the activity of the promoter region can subsequently be deduced from the expression level of the reporter genes. However, the DNA sequences in question, e.g. the promoter regions are detached from their endogenous context [10–15]. Thus, the experiments must be performed outside the natural environment of the promoter. Furthermore, classical promoter studies assume an independent effect of *cis*-regulatory regions which is reflected in their experimental setup. In conclusion, it is not possible to study complex interactions of individual regulatory elements.

Focusing on human renin, a "*renin* enhancer" (about -12.000 base pairs (bp) to transcription start site (TSS)) as part of an evolutionary conserved region (hRENc region), which is considered to be important for basal *REN* expression [11, 16, 17]; and more proximally, a "chorion enhancer" (about -5,500 bp to TSS) was found. However, its relevance is still unclear [18].

The closest known regulatory region to transcriptional initiation is the proximal *renin* promoter, which has been shown to play a significant role in tissue and cell specificity of *REN* expression following experiments on transgenic mice [17, 19]. Research on the As4.1 cell line has identified a proximal promoter region of the murine *Ren-1C*, for which a position in the human *REN* is usually indicated at about -200 to +6 upstream of the transcription start site [8–10]. This region shows distinctive homologies between mice and humans up to a fully conserved TATA box [8]. It´s considered to be essential for tissue-specific expression of *REN*, even though the proximal promoter region has reached only a slight enhancement of *REN* expression in murine reporter assays [11, 14]. However, the renal enhancer of *REN* has a lower *trans*-activating capacity compared to the murine renal enhancer [20], which could enhance the influence of the proximal promoter in the *REN*.

Reporter studies and Electrophoretic Mobility-Shift Assays (EMSA) have identified numerous transcription factor binding sites for the proximal promoter region. So, there is e.g. evidence for binding sites that are important for gene regulation of *REN* via the second messenger cAMP [21–23]. Thus, regulatory elements were also found, whose relevance to transcriptional regulation of *REN* seems to be questionable [24–26].

In addition to the *cis*-regulatory elements, the transcription factors (TFs) are essential to enable gene transcription. These proteins bind to DNA and can activate or repress the transcription of genes. There are differences in the way TFs act to regulate gene expression. Some TFs need to assemble with other proteins, others can directly recruit RNA polymerase which then leads to gene transcription [27]. In a current review, a distinction is made between approx. 1600 human TFs, which represent ~8% of all human genes [28]. There are several ways to classify TFs. In general, a division into basal or general TFs and specific TFs is possible. Basal TFs are ubiquitous in all cells and necessary for transcription to occur [29]. During assembly they are part of the preinitiation complex that enables the binding of the RNA polymerase and thus the initiation of the transcription via specific DNA binding sites such as TATA boxes [29, 30]. In contrast, specific TFs only show activity in specific tissues and/or at specific developmental stages. They may bind at specific DNA binding sites (*cis*-regulatory regions), e.g. promoters, enhancers or silencers and are necessary for the regulation of central mechanisms such as cell development or the response to stimuli via signal cascades [31, 32].

A novel experimental approach to explore and thus better understand the complex mechanisms of transcriptional regulation via *cis*-regulatory elements has become possible through further development of the CRISPR-Cas9 system.

This system has become a powerful gene editing engine. It facilitated and expanded the possibilities of loss-of-function and also gain-of-function studies and should even be highly valuable for studying circadian rhythms [33].

Apart from gene editing the guided binding of the Cas9-RNA-DNA complex can be used for other purposes. This includes gene activation, gene silencing and gene labelling, for example with fluorescent proteins. In this study we used a modified Cas9 system that is able to activate genes. Therefore, a non-cutting Cas9, that was fused to an activation transcription factor complex was chosen. We used this system to explore the *cis*-regulatory importance of genomic DNA, in this study for the proximal *renin* promoter.

In detail, we used the Synergistic activation mediator complex (SAM) [34] (Fig 1).

In 2015, Konermann et al. presented a modified Cas9 (dCas9) that was coupled to a complex of different transcription factors, allowing them to establish a highly efficient guide-RNA-directed *trans*-activating complex referred to as SAM complex [34–38]. The dCas9 lost its originally cutting function through mutations in its catalytic domain, making it an efficient DNA-binding protein [39]. Additionally, it is coupled to the transcription factor VP64 [40, 41]. Regions of the original tracr-RNA were also modified to bind a complex of trans-activating domains of p65 and HSF1 [34]. The dCas9 can be programmed by a single guide-RNA [39]. Regarding the design of the guide-RNA it is possible to target any sequence in the genome, given that a PAM site (NGG) follows the 20 nt sequence. In order to make the interactions specific, the sequence should be unique for the given species. This is incorporated by the bioinformatic tools for designing potential target sequences.

The development of this new gene editing tool enables the study of genes and their regulatory elements in their endogenous context and in their natural environment. This kind of research is not possible by classical reporter studies (as described above). In addition, this specific and constant *trans*-factor complex allows the isolated study of the *cis*-regulatory regions since the same trans-factors always bind to the regulatory elements.

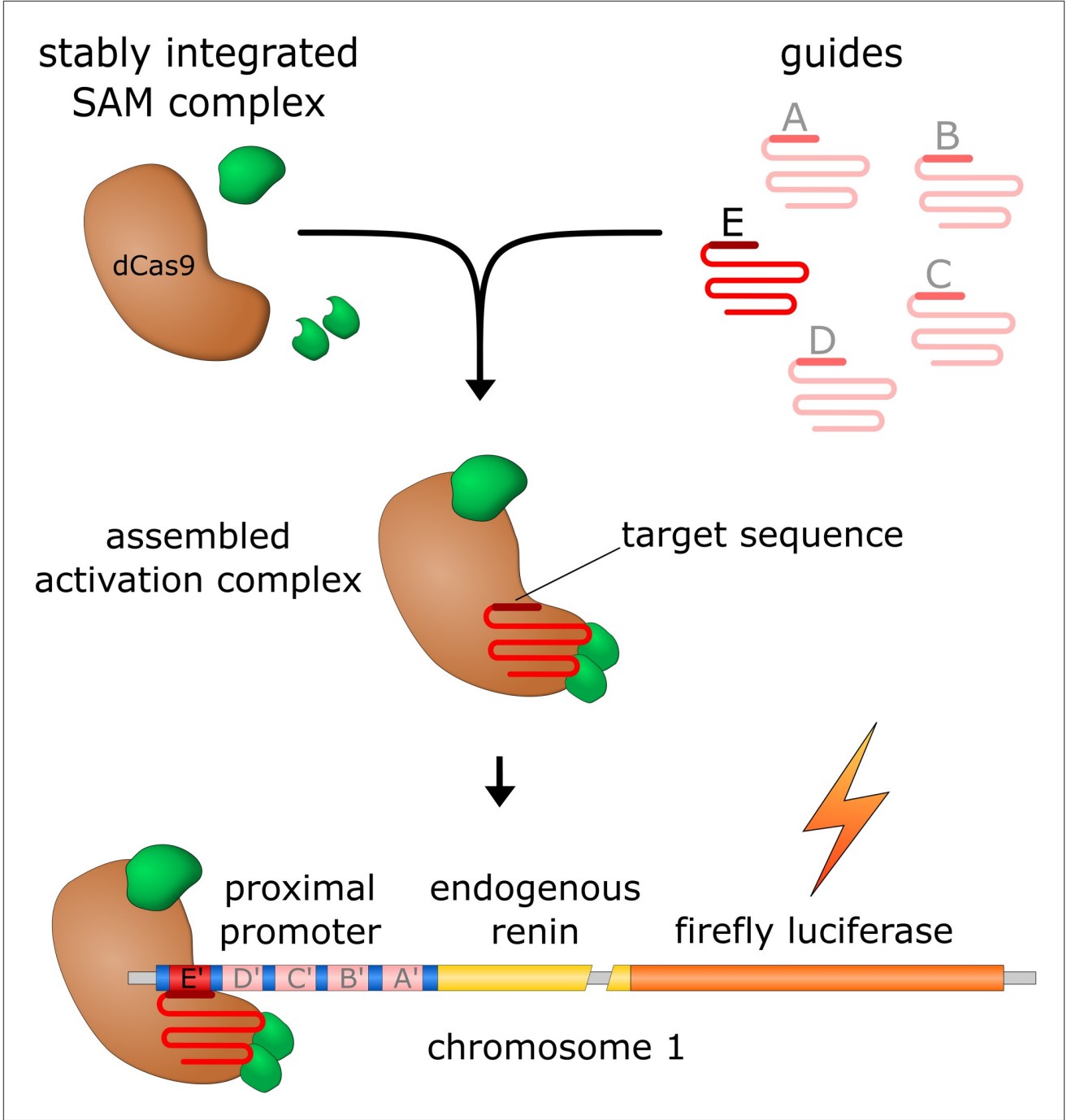

**Fig 1. Programmable site-specific activation of REN with designed guides A-E.** After assembling of the activation complex (SAM) the attaching guide (shown here by the example of guide E) leads the complex to its specific region of the proximal renin promoter (E'). Thus, the guides enable site specific programming of the SAM complex. The resulting gene activation can be measured by a tagged firefly luciferase.

In order to obtain more understanding about the complex transcriptional network, mathematical modelling of experimental data was performed. For example, conforming principles have been identified, under which individual sections of a *cis*-regulatory element approximately 14 kb upstream of *REN* interact with each other [42]. The possibility of drawing

conclusions concerning the dynamics of *cis*-regulatory elements by using mathematical modelling was already shown in 2003 in a study on the *Escherichia coli* bacterium [43]. In addition, by mathematical modelling in 1998, complex processes at an important promoter region of *Endo1* had been revealed, which are of great importance for the embryonic development of sea urchins [1]. Thus, mathematical modelling can help to deepen the understanding of the complex transcriptional network.

When statistically examining factors that influence a variable, it is of general scientific interest to find out whether these factors act independently or interdependently. In our opinion, this question of complex interactions was given too little importance in the research on *cis*-regulatory regions of *REN*, especially on the proximal promoter [9, 26]. A study that attempted to address this problem with a modelling approach emerged in 2007 [42]. However, even here the experiments were plasmid-based and therefore outside of the endogenous context. In addition, a regulatory region of *REN* was in the focus, which is approximately 14.000 base pairs upstream of the start of transcription.

With this study on the one hand we want to show a novel approach to investigate interactions of *cis*-regulatory regions in an endogenous context, and on the other hand raise awareness of how important these interactions are for the understanding of the complex transcriptional network.

To study possible complex interactions within the proximal promoter region of *REN* we applied a novel combined approach. Firstly, this approach consists of combinatorial transfections from five selected guide-RNAs that translocate the SAM-complex to a specific region of the endogenous proximal promoter. The resulting expression levels of *REN* through the different combinations of the targeted promoter regions can be quantified by luciferase activity. Secondly, we generated and fitted two different mathematical models to our experimental data. When modelling experimental data, the principle of simplicity should always be considered. The simplest assumption would be that the regions examined influence the expression of *REN* completely independent of each other. This is reflected in our first model (sum model). The sum model describes an independent relationship of the promoter sequences we examined to their influence on the activation of *REN*. In this model each individual region is analysed in respect to its influence on gene expression. This approach is comparable to the knowledge that classical promoter studies can provide, since, as described above, these do not allow the possibility of studying complex interactions. However, an important scientific question in modelling is whether a statistical interaction exists [44]. In order to address this issue, a second mathematical model was generated (interaction model). This model represents a more complex assumption of the conditions in the region of the proximal promoter. The interaction model moreover allows interactions between the selected promoter regions to explain the *REN* activation. Regarding modelling we used the multiple linear regression model to fit the linear parameters, which is a standard statistical method [45]. In order to check which of the generated models can explain the measured data best, the respective absolute prediction error was calculated. Following the principle of maximum parsimony in modelling, the respective Akaike information criterion (AIC) was calculated for further model judging [46]. The objective of this study was to examine potential interactions of sequences within the proximal *renin* promoter. Through combinatorial transfections of specific guide-RNAs using the SAM complex and computational modelling of the measured data we want to show a novel combined approach that helps to enlighten the complexity of *cis*-regulatory regions in an endogenous context. We want to show that transcriptional regulation is even more complex than already known and that complex interactions should be considered when assessing the importance of specific *cis*-regulatory elements. The region of the proximal *renin* promoter is in the focus of this work. Thus, this study may help to better understand the transcriptional regulation of the

key enzyme of RAAS. Furthermore, in our opinion this approach is also suitable for evaluating interactions and dynamics of other *cis*-regulatory regions.

## Material and methods

### Cell line

Human embryonic kidney cells (HEK293) were cultured in T75 cell culture flasks in high-glucose DMEM (Thermo) supplemented with 10% fetal bovine serum (FBS) (Biochrom) and 1% penicillin/streptomycin (Biochrom) at 37˚C and 5% $CO_2$ in a humidified incubator. The medium was changed every 3–4 days. At approximately 90% confluence, the cells were passaged at a 1: 5 dilution and seeded in a new T75 flask.

### Generating SAM-complex expressing cells by lentiviral transduction

24 hours prior to transfection, $1.4*10^6$ cells were seeded into cell culture dishes (60 mm diameter) in 3 ml high-glucose DMEM supplemented with 10% fetal bovine serum. Transfection was performed at a confluency of 80–90%. Per dish 2 μg of either lenti dCAS-VP64_Blast (Addgene #61425) or lenti MS2-P65-HSF1_Hygro (Addgene #61426) [34], and the lentiviral packaging plasmids 2 μg pMDLg/PRRE (Addgene #12251) [47], 0.4 μg pRSVrev (Addgene #12253) [47] and pMD2.G (Addgene #12259) were diluted in BES-buffer (50 mM BES, 280 mM NaCl, 1.5 mM $Na_2HPO_4$ x 2 $H_2O$, pH 6.95). $CaCl_2$ was added to a final concentration of 0.125 M, the plasmids were incubated for 10 minutes at room temperature and added to the cells. Post transfection (12–16 hours later), the medium was changed to fresh DMEM with 10% FBS. Lentivirus-containing supernatant was harvested 48 hours after transfection. The supernatant was filtered sterile with a 0.45 μm cellulose acetate filter (Roth) and stored at -80˚C.

Approximately 50.000 HEK293 cells were seeded per well of a 24-well plate in 500 μl high-glucose DMEM supplemented with 10% fetal bovine serum. The medium was changed to 500 μl of lentivirus supernatant after 12 and 18 hours. First, cells were generated expressing dCas9-VP64; afterwards, MS2-P65-HSF1 were selected with 10 μgml$^{-1}$ blasticidin (Invivogen) and 100 μgml$^{-1}$ hygromycin B (Invitrogen). Cells are now referred to as HEK dCas9-SAM.

### Tagging of *REN*

*REN* was tagged in frame with the gene for firefly luciferase (Fig 1) and the G418 resistance using a Cas9 and a specially designed guide-RNA GGCTTCGCCTTGGCCCGCTG. The G418 resistance cassette was not used in this study but was cloned for potential further experiments with those cells or plasmids. The guide design was performed by the guide design tool "crispr. mit.edu" and cloned according to the manufacturer's instructions into pGuide-it-tdTomato (Clontech). The stop codon was deleted. Luciferase and G418 resistance were linked to the *REN* via T2A and P2A sequences. The flanking homology arms were amplified from genomic DNA of HEK293 cells by PCR. The following primers were used for the reaction:

Renin-left-arm-forward-NotI GTACGCGGCCGCCGCTCACCAGCGCGGACTATGTAT,
Renin-left-arm-reverse-PacI AGCTTTAATTAAGCGGGCCAAGGCGAAGCCAATGCG,
Renin-right-arm-forward-AarI acgtccacctgcgtgcttaaaggccctctgccacccag gcag,
Renin-right-arm-reverse-AscI AGCTGGCGCGCCGACCCAAGTCAGACGGGCTGGGTTC.
The homology arm PCR products were integrated into MV-PGK-Puro-TK vector (Transposagen), which was modified by integrating a cassette via NheI and HindIII digestion. The cassette contains a P2A-linker-firefly luciferase-T2A linker-G418 resistance-PGK promotor-

puromycin resistance-T2A linker-thymidine kinase (S1 Fig) and is flanked by restriction sites NotI and PacI for left homology arm integration and AarI and AscI for right homology arm integration.

For transfection of the plasmids $1.2^*10^6$ HEK dCas9-SAM were seeded into a well of a 6-well plate and transfected 12–16 hours later at a confluency of 80–90%. 1.5 μg of *REN*-tagging donor plasmid and 1.5 μg of pGuide-it-tdTomato vector with the integrated *REN*-guide were diluted in 100 μl Opti-MEM™ (Thermo) and mixed with 100 μl Opti-MEM™ including 12 μl Lipofectamine® 2000 (Thermo). After 5 minutes incubation at room temperature, the plasmid was added dropwise to the cells. Cells were selected with 2 μgml⁻¹ puromycin. The cells are now referred to as HEK dCas9-SAM_Renin-luciferase.

### *REN-guides*: Design and cloning

The guides used for this work were designed using the online "SAM sgRNA design tool" [48] in the beginning of 2018. The guides are sorted in order of specificity (highest to lowest) based on a method described by Hsu et al in 2014 [49]. We have chosen the top 5 hits for human *REN* for our experimental approach. The respective 20-base-long promoter sections A' to E' are approximately evenly distributed from 60 to 159 base pairs upstream of the transcription start of *REN* (Table 1).

The DNA oligos were designed with overhangs for BbsI and cloned into backbone plasmid sgRNA (MS2) ordered from Addgene (#61424) [34] following the depositor's advice. Thus, guides A-E were created.

### Transfection of the guides

Approximately 50,000 HEK_dCas9-SAM_Renin-luciferase cells were seeded per well in 96-well plates in 140 μl each of high-glucose DMEM containing 10% FBS, 1% Pen/Strep, 1 mM L-Glutamin (Biochrom), 10 mM HEPES (Thermo) and 250 μM Luciferin D (Promega). The cells were incubated overnight in a cabinet humidified incubator at 37°C and 5% $CO_2$. After 16 hours, the transfection with the guides was carried out at a confluence of about 80%. The transfection was performed according to the protocol of Lipofectamine® 2000 reagent. 0.5 μl Lipofectamine® 2000 and a constant amount of guide-DNA per guide and per well were added in a total volume of 10 μl Opti-MEM™.

To study whether interactions between the investigated promoter sequences occur all possible combinations of the five guides A-E were transfected into the HEK_dCas9-SAM_Renin-luciferase cells. In each case 30 ng of guide-DNA per guide and per well were used, which means the total amount of DNA per well varied from 0 ng– 150 ng. Each possible combination was transfected with a sample size of n = 6. We have not performed a concentration dependent

**Table 1. Chosen guides A to E for combinatorial transfections.**

| Guide | Sequence (5'-3') | Distance to TSS |
|---|---|---|
| A | TCTGCCCTGATTTATTACCC | -60 bis -80 |
| B | ACTGCCCTGCCATCTACCCC | -81 bis -101 |
| C | TCTCACTGCGGGACAGAGCT | -111 bis -131 |
| D | GTGAAGGGTACCCAGGTTTC | -137 bis -157 |
| E | CAGCCCCTCTGCTCCCCATC | -159 bis -179 |

Here the five chosen guides A to E with their respective distance to TSS are shown that are used for the experimental work. For guide-design we have used the online "SAM sgRNA design tool" in the beginning of 2018.

**Table 2. Pattern of the combinatorial transfections of the designed guides A-E.**

|     | A | B | C | D | E |
| --- | --- | --- | --- | --- | --- |
| 1   |   |   |   |   |   |
| 2   |   |   |   |   | X |
| 3   |   |   |   | X |   |
| 4   |   |   |   | X | X |
| 5   |   |   | X |   |   |
| 6   |   |   | X |   | X |
| 7   |   |   | X | X |   |
| 8   |   |   | X | X | X |
| 9   |   | X |   |   |   |
| 10  |   | X |   |   | X |
| 11  |   | X |   | X |   |
| 12  |   | X |   | X | X |
| 13  |   | X | X |   |   |
| 14  |   | X | X |   | X |
| 15  |   | X | X | X |   |
| 16  |   | X | X | X | X |
| 17  | X |   |   |   |   |
| 18  | X |   |   |   | X |
| 19  | X |   |   | X |   |
| 20  | X |   |   | X | X |
| 21  | X |   | X |   |   |
| 22  | X |   | X |   | X |
| 23  | X |   | X | X |   |
| 24  | X |   | X | X | X |
| 25  | X | X |   |   |   |
| 26  | X | X |   |   | X |
| 27  | X | X |   | X |   |
| 28  | X | X |   | X | X |
| 29  | X | X | X |   |   |
| 30  | X | X | X |   | X |
| 31  | X | X | X | X |   |
| 32  | X | X | X | X | X |

All performed combinatorial transfections of the designed guides A-E. To sort the combinations, a binary code was applied so that each combination could be uniquely assigned.

analysis for each individual guide in this study. From other studies of other genes in our lab we have used similar concentrations for transfection of guide-RNAs that showed biologically relevant expression of RNA and protein. For this reason, we always used the concentration of 30 ng guide-RNA per well for transfection in the combinatorial setup.

In order to sort the combinations, a binary code was applied so that each combination could be uniquely assigned (Table 2).

## Luciferase-reporter-assay

The luciferase assays were performed with HEK_dCas9-SAM_Renin-luciferase cells. The cells were transfected with all possible combinations of the five guides as described above. The sample size of each combination was n = 6. Immediately after transfection, the 96-well plates were

sealed and the luciferase activity was measured over time at 37°C in the TopCount® NXT (Perkin Elmer, Waltham, USA). Luciferase activity is expressed in counts per second (cps).

## Modelling of received data and statistics

In order to find conforming patterns by which the promoter regions influence the expression of *REN*, models were generated with "R". The first model (sum model) describes an independent relationship of the five selected promoter sequences A'-E' with respect to their influence on the activation of *REN*. This model was chosen because it describes the simplest possible form of framework of the regulatory elements. In this model each individual region is analysed concerning its influence on gene expression. For the independent sum model, the measured activity $y_i$ can be represented by formula 1:

$$y_i = \in_i + \beta_0 + A_i\beta_1 + B_i\beta_2 + C_i\beta_3 + D_i\beta_4 + E_i\beta_5$$

where $\beta_0$ is the offset, $\beta_{1...5}$ are the linear coefficients, $A_i...E_i$ are {0,1} depending on presence in experiment $i$ and $\in_i$ is the error.

Furthermore, a second model has been generated (interaction model) that additionally allows interactions between the promoter sequences in order to explain *REN* activation. For the interaction model the measured activity $y_i$ for all the 32 possible combinations can be represented by formula 2:

$$\begin{aligned} y_i = {} & \in_i + \beta_0 + \beta_1 A_i + \beta_2 B_i + \beta_3 C_i + \beta_4 D_i + \beta_5 E_i + \beta_6 A_i B_i + \beta_7 A_i C_i + \beta_8 B_i C_i + \beta_9 A_i D_i \\ & + \beta_{10} B_i D_i + \beta_{11} C_i D_i + \beta_{12} A_i E_i + \beta_{13} B_i E_i + \beta_{14} C_i E_i + \beta_{15} D_i E_i + \beta_{16} A_i B_i C_i + \beta_{17} A_i B_i D_i \\ & + \beta_{18} A_i C_i D_i + \beta_{19} B_i C_i D_i + \beta_{20} A_i B_i E_i + \beta_{21} A_i C_i E_i + \beta_{22} B_i C_i E_i + \beta_{23} A_i D_i E_i + \beta_{24} B_i D_i E_i \\ & + \beta_{25} C_i D_i E_i + \beta_{26} A_i B_i C_i D_i + \beta_{27} A_i B_i C_i E_i + \beta_{28} A_i B_i D_i E_i + \beta_{29} A_i C_i D_i E_i + \beta_{30} B_i C_i D_i E_i \\ & + \beta_{31} A_i B_i C_i D_i E_i \end{aligned}$$

where $\beta_0$ is the offset, $\beta_{1...31}$ are the linear coefficients, $A_i...E_i$ are {0,1} depending on presence in experiment $i$ and $\in_i$ is the error.

These models have been fitted with the measured data. For fitting of the models represented in formula 1 and 2 the fitting functionality of the *lm* function of the built-in *stats* package of R version 1.1.423 was used [50]. The idea behind this is to minimize the error in prediction of *y* by optimizing the linear factors *β* for the experimental data. The *lm* function of the R package *stats* computes the linear factors *β* that fit the input variables A to E according to the proposed model including all the statistics of the fitted parameters. The complete data that was used including the R-script can be found in the supplementary material (S1 Table, S1 Script). Data was analysed at 60 hours after transfection.

For the evaluation of the models we used the obtained coefficients of the respective model and put the coefficients in formula 1 and 2, respectively. For statistical analysis of the coefficients itself the built-in p-value calculation of the multiple linear regression of the *lm* function of R was used. For comparison of the models we calculated the respective absolute prediction errors of the two models with the built-in *predict* function of the stats package of R [50]. After fitting of the linear factors *β*, the error $\in_i$ could be calculated for the experiments *i* through conversion of the respective formula. For statistical analysis of the respective absolute prediction errors we used the independent 2-group Mann-Whitney U Test [50]. To compare the generated models following the principle of maximum parsimony in modelling, the *AIC* (Akaike Information Criterion) of each model were calculated for further model judging. The *AIC* was calculated in the R environment with the *AIC* function of the *stats* package [50].

Random controls were generated by the built-in *sample* function of R [50]. For this purpose, a dataset was created by randomizing the values of the respective variables A to E. This

randomized data was then fitted to model 1 and 2. No statistically significant p values for the coefficients were obtained in the random case (S1 Script).

The entire statistical evaluation including model fitting and model evaluation by 2-group Mann-Whitney U Test of the respective prediction errors and calculation of the AIC did not need any further parameters, apart from the experimental parameters such as incubation temperature or evaluation at about 60 hours after transfection.

## Results

Our aim was to explore possible interactions within the proximal *renin* promoter. Therefore, HEK cells were transfected with all combinations of guides that represent the selected promoter sequences A' to E'. The luciferase signal of transfected HEK dCas9-SAM_Renin-luciferase cells, that indicates renin activation, was measured over time in the TopCount® NXT. With a modified Venn diagram the activation levels of *REN* via all of the 32 resulting possible combinations of the guides were visualised (Fig 2). The 32 individual areas result from overlaps of the five main areas A-E, which represent the guides. The size of the respective area is of no significance considering the results. The circle area surrounding the entire individual areas reflects the case in which none of the guides were used.

Each combination of guide RNAs has its own binary code assigned. Each transfection of HEK dCas9-SAM_Renin-luciferase was done according to the same protocol (see Material and methods).

After approximately 24 hours of transfection of the combinations of RNA guides to HEK dCas9-SAM_Renin-luciferase, expression of the *REN* could already be detected via the increase in the luciferase activity. Over time, the expression levels differed for the individual combinations of the promoter sequences (Fig 3). In Fig 3 expression levels of the 32 possible combinations are shown at three different time points after transfection using the modified Venn diagram as described above (Fig 2). Respective activation levels of *REN* were expressed in a colour ramp that rose in ascending order from white to yellow, further to red and up to blue. For analysis, the values of expression levels were scaled logarithmically. Otherwise, since

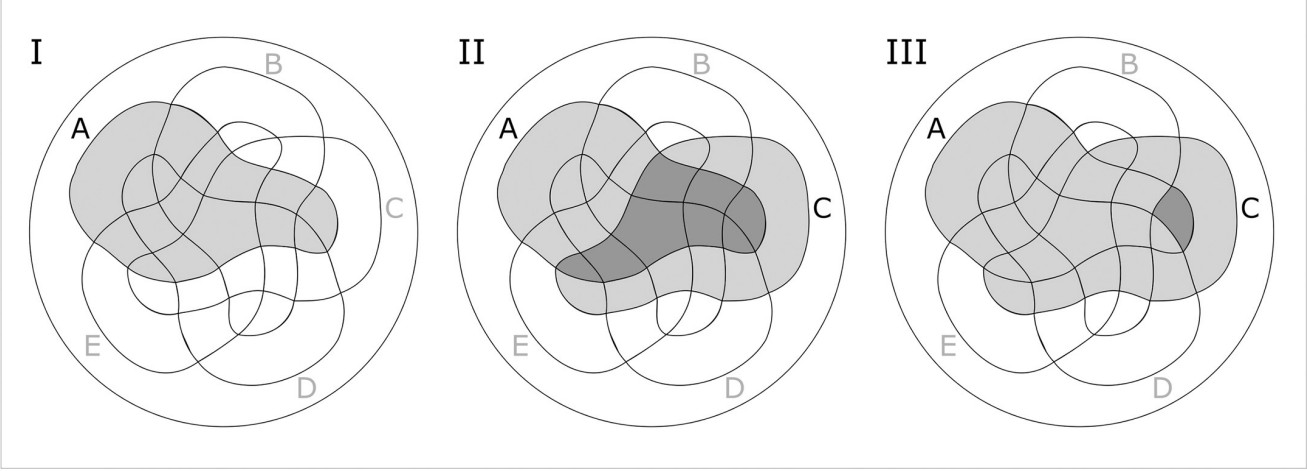

**Fig 2. Visualisation of the 32 possible combinations of guides.** This modified Venn diagram represents each of the 32 possible combinations of guides as a surface. The individual areas result from overlaps of the five main areas A-E, which represent the guides. The size of the respective area is of no significance considering the results. The circle area surrounding the entire individual areas reflects the case in which none of the guides were used. Three examples are shown here. (I) All combinations including A are filled light grey; (II) All combinations including A and C are filled in dark grey; (III) Combination consisting solely of A and C are marked dark grey.

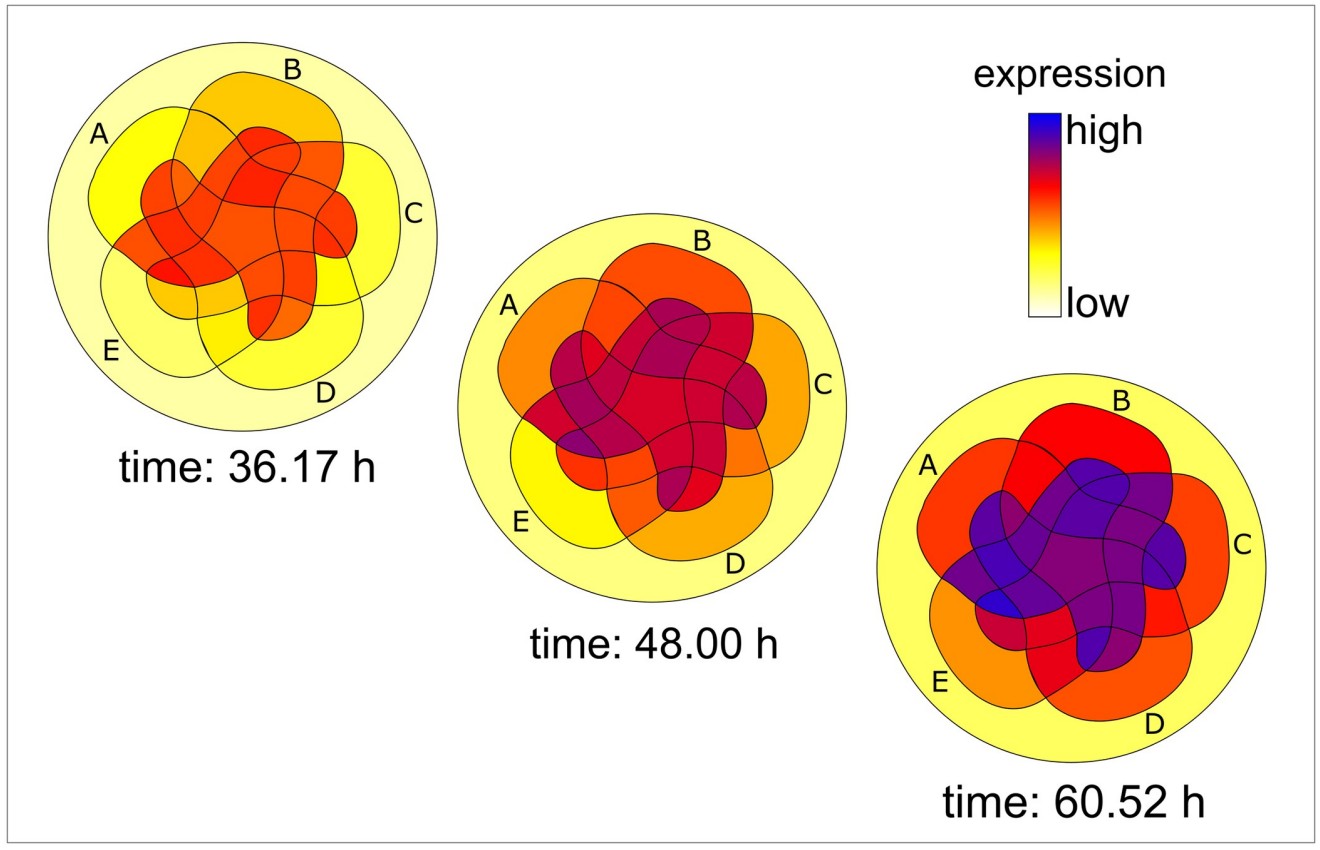

**Fig 3. Visualisation of the luminescence shows the activation of renin via cis-regulatory regions for each individual combination of guides over time.** The expression levels of the 32 possible combinations are shown here at three different time points after transfection of HEK dCas9-SAM_Renin-luciferase. By translating the luminescence values into a colour scale, the corresponding colour intensity resulted for each RNA guide combination. A time-dependent increase in the expression levels of the 32 guide combinations becomes obvious. Furthermore, individual guides and the 32 possible combinations caused different expression levels of REN. The axis "Expression" is scaled logarithmically (each area n = 6).

these values were widely divergent, only a few surfaces would have been coloured in a linear scale, whereas most were white to slightly yellow. On the one hand the individual guides itself caused different levels of renin expression. On the other hand, the different combinations achieved different renin activation (Fig 3 and S1 Table). The combinations of the two guides C—B or A—E caused a renin activation that was higher than the simple summation of the respective individual activation levels. In contrast, combinations of the two guides A—B or also D—C have a less increasing effect on the renin activation.

To further investigate the potential interactions of the chosen proximal promoter sequences we generated two mathematical models in R and fitted them to the measured data described above.

The first model was the sum model. It describes an independent relationship of the five selected promoter sequences with respect to their influence on activation of *REN*. We chose this model because it assumes the simplest possible correlation between the promoter regions. Modelling of the data revealed that all sequences studied significantly affected the activation of *REN* (S2 Fig). However, the assessed values of the 32 combinations often show deviations to the predictions of the sum model (Fig 4). The measured luminescence values were smaller than the values predicted by the sum model after transfection of all combinations in which neither guide A nor B were involved. As soon as either A or B was present in the combinations,

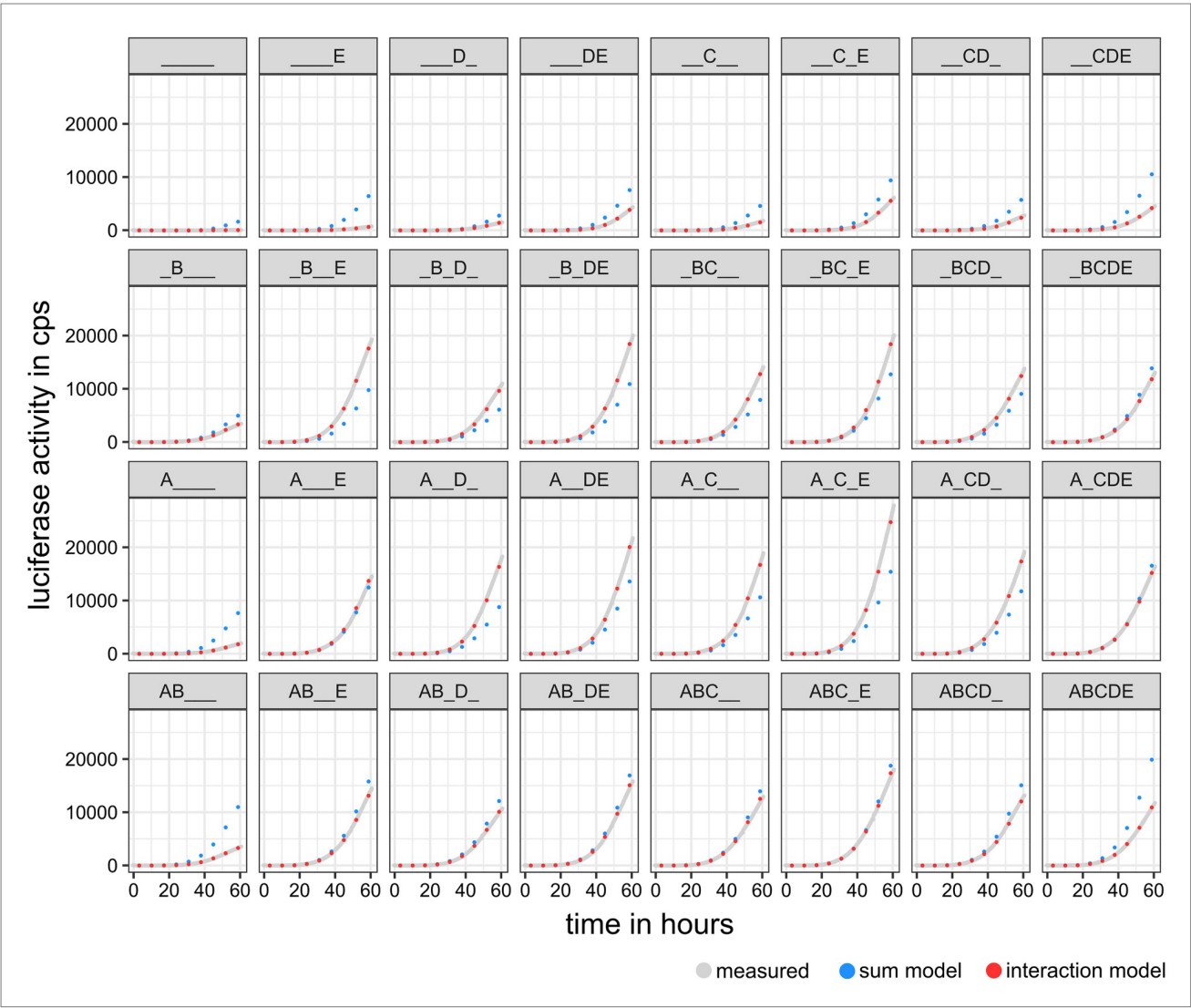

**Fig 4. Luciferase measurements vs. predicted values of the models.** The measured values (grey) of each combination are plotted with the respective prediction of the **sum model** (blue) and the **interaction model** (red). The measured values show deviations from the predicted values of the **sum model**. The prediction of the **interaction model** better matches the measured expression levels of the 32 different combinations. The luciferase activity is expressed in counts per second (cps). The measured values are presented by their medians. The curves of the measured values result from >3,000 measuring points per combination (32 combinations, each n = 6).

the luciferase activities were higher or at least at the level of the prediction. Exceptions were combinations in which A or B stimulated the expression of the *REN* in each case by solely driving the promoter sequence. Interestingly, when guides A and B were both present in the combinations, values in the range of the prediction or below were measured.

The second investigated model, the interaction model, allows interactions existing between the individual promoter regions in order to explain the measured data. The deviations of the predicted values of the interaction model are smaller than the predicted values of the sum model (Fig 4). Thus, the interaction model seems to better describe the measured expression levels of the 32 different guide combinations.

In the next step we compared the two models. By calculating differences of the measured values against the predicted values of the two models, we were able to visualise the degree of

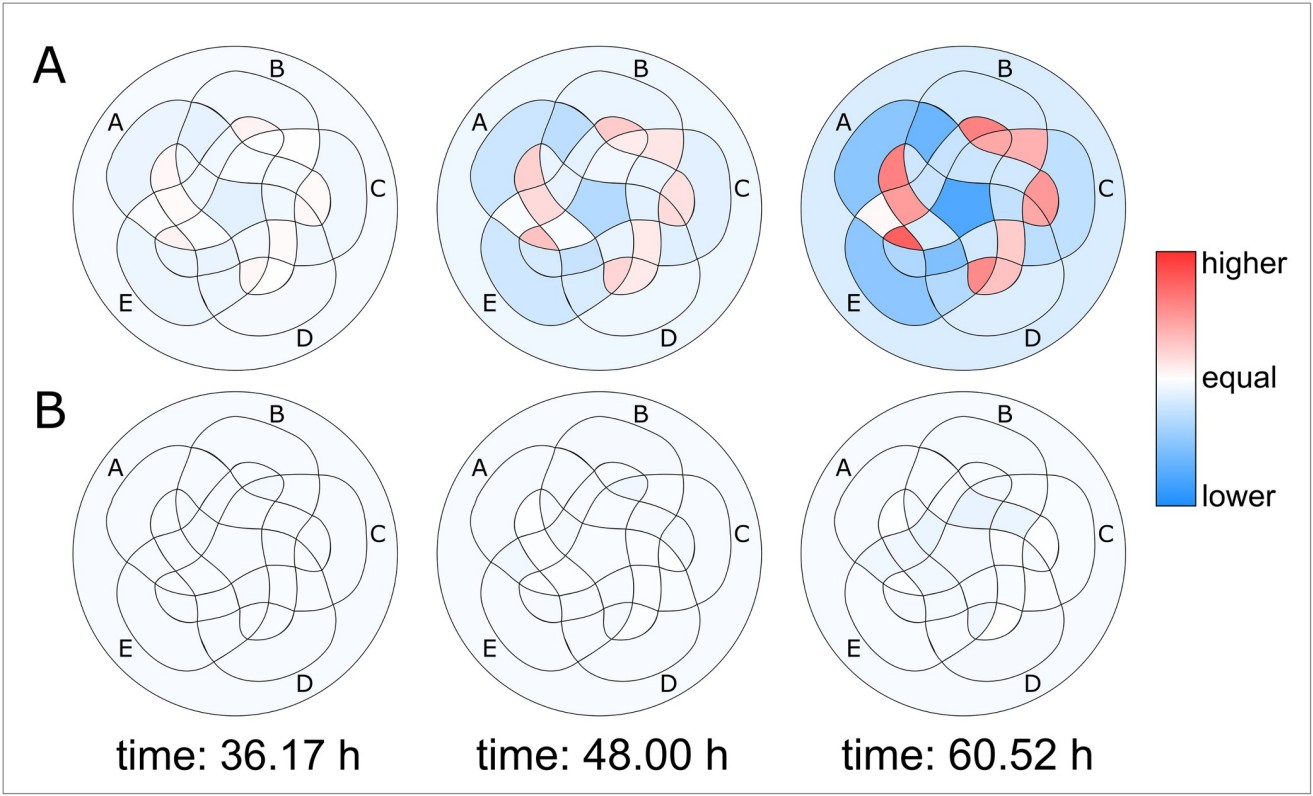

**Fig 5. Differences between measured values and predicted values of the models at three different time points.** The differences of measured values and predicted values of the 32 possible combinations assuming the two generated models are shown here for three different time points after transfection. Areas turn red when the differences between the measured luciferase activity and the predicted values of the respective model are positive and turn blue when the differences are negative. (A) In the sum model over time, the colouration of the combinations increases. (B) For the interaction model almost no colouration can be recognised over time. The values of the differences are linearly scaled (each area n = 6).

similarity at three chosen time points at Fig 5. It becomes obvious that an independent relationship of the promoter sequences as assumed in the sum model cannot adequately explain *REN* expression.

The prediction of the sum model and the measured values differ, as seen at Figs 4 and 5. This motivated us to further analyse the better fitting interaction model. The influences of the linear single factors A to E are no longer significant (with exception of B) when interaction of the sequences is allowed (Fig 6). The activities of the combinations A—C, A—D, A—E, B—C, B—D, B—E significantly outweighed those linear single factors. These results underline that interactions within the promoter regions are essential for the extent of the assessed gene activity.

Further, for both models, the absolute error (absolute difference between measured and predicted model values) was calculated. The error was significantly smaller for the interaction model than for the sum model ($p < 2.2*10^{-16}$ according to 2-group Mann-Whitney U Test; Fig 7). For further comparison, the AIC was calculated for both models. This AIC calculation returned a value of 3881.4 for the sum model and a value of 3609.6 for the interaction model, indicating the interaction model to be statistically more adequate. In order to check whether these results could have been a coincidence, all measurement data of the individual promoter sequences were randomised. After applying the generated models to this randomised data, no significant effects on the expression of *REN* were found for the selected promoter sequences (S1 Script).

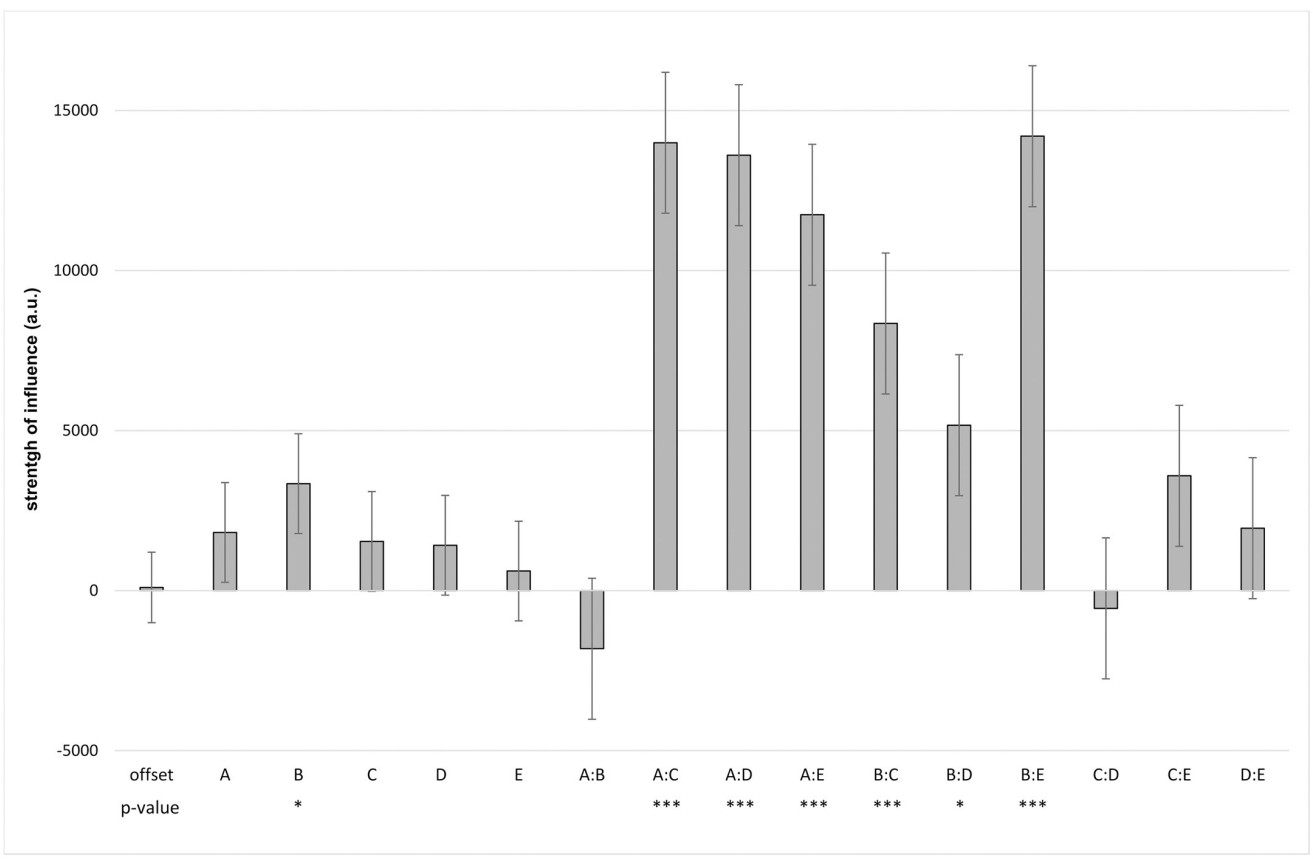

**Fig 6. Influence of the promoter regions (A to E) and their potential interactions on gene activity of REN.** The strength of influence of certain combinations of promoter sequences outweighs the influences of the singular linear individual factors. Influences of linear single factors A-E are no longer significant, with exception of B. Significances resulted from the multiple linear regression modelling with the lm function of the build in stats package in R. Strength of influence is represented as arbitrary unit (a.u.). (mean and SD, n = 6, p-values: * <0.05, *** <0.0005).

## Discussion

In a combined approach, consisting of combinatorial transfections and computational modelling of the measured data we analysed the role of *cis*-regulatory elements in the proximal promoter of human *REN*. According to research on murine *Ren-1C* via reporter assays, the proximal *renin* promoter is described to have low activation potential for the *REN* [11, 14]. However, compared to the murine renal enhancer the gene activating capacity of human renal enhancer is lower [20], which could enhance the influence of the proximal promoter in the *REN*. Furthermore, a 99% decrease in the transcriptional activity of renin was also found when the region of the proximal promoter was deleted in reporter assays performed in As4.1 cells [26].

However, in our experimental setup the *REN* expression could be triggered by the targeted translocation of the SAM complexes to all five promoter sequences (Fig 3, S2 Fig).

The combinatorial transfections were performed with a constant amount of guide-DNA per guide. In a previously unpublished study, we have determined renin levels after activation of this promoter region. We were able to show that the catalytic activity of renin was present in the supernatant of the HEK cells. This shows that we operate in a biological sensible range of renin expression. Further it shows that secreted renin (biologically relevant for blood pressure regulation) was produced in our cells.

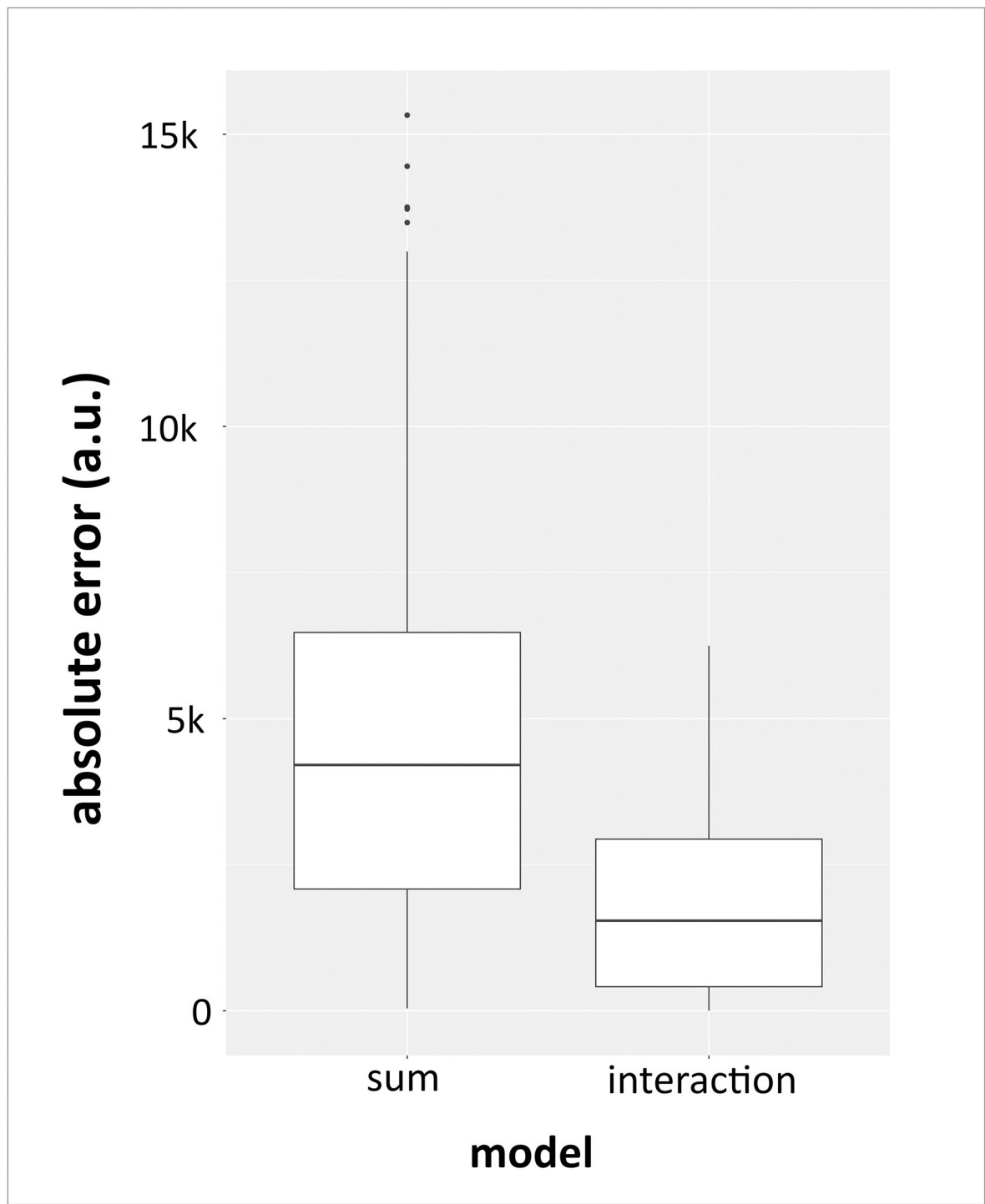

**Fig 7. Distribution of the respective absolute error of the sum model vs. the interaction model.** Here, the two models are compared by calculation of their respective absolute prediction errors. The distribution of the absolute errors is shown as a box plot. The absolute prediction error of the interaction model is significantly smaller than that of the sum model (p<2.2$^*10^{-16}$ according to 2-group Mann-Whitney U Test). Absolute error is represented as arbitrary unit (a.u.).

Our new approach includes computational modelling of the measured data besides the combinatorial transfections. This enabled deeper insights into *cis*-regulatory interactions. Independent linear models are a method to describe multifactorial influences on an output variable (in this case renin transcription). In the simplest case this would be a two-factor setup, e.g. whether a drug (factor 1) acts dependent on sex (factor 2). An example of a linear sum model would be the calculation of the melting temperature ($T_m$) of the DNA: $T_m = 4(G + C) + 2(A + T)$. A more advanced and complex formula that considers the neighbouring relationships of the base pairs was described by Hooyberghs et al. in 2009 [51].

Considering the independent **sum model**, guide A reached the most potent effect. The homologous region coincides with a well conserved sequence in mice and humans and is considered to be an important sequence for responding to cAMP stimulation within the proximal *renin* promoter [21]. cAMP is known to be an important transcription factor for stimulating *REN* transcription [5, 9].

So far, we can state that the examined promoter regions seem to have different degrees of influence on gene expression of *REN*. The different measured expression levels of the individual transfections can depend not only on the possible different activation ability but also on other factors that were not considered in this work. For example, the regions examined can be occupied by other TFs. A special secondary structure of the DNA or the arrangement of the nucleosomes might also make the access to the SAM complex more difficult [52]. However, this gain in knowledge would probably also have been possible with classical reporter assays. Most of the knowledge about the importance of *cis*-regulatory areas of the *REN* is based on this type of promoter studies [8, 9].

However, whether there are interactions within *cis*-regulatory regions should be a central question in the research of the complex transcriptional network [44]. In particular, if one looks at the proximal promoter of *REN*, in our opinion, the question about the existence of complex interactions and whether these may have significance for the interpretation of *cis*-regulatory elements was given too little attention [9, 26]. Due to the experimental setup, the ability to explore these dynamics is a limitation for classical promoter assays. With our approach, however, this question could be examined.

To address this issue, we enabled more complexity to explain the measured expression levels. The next level of complexity would include—apart from the linear scaling factors–those factors describing statistical interaction. Therefore, an **interaction model** was subsequently computed and fitted to the measured data. Looking at the interaction model we could see the importance of interaction (Fig 6). By allowing interactions of the promoter regions the influences of the linear single factors A to E were no longer significant (with exception of B). Certain sequence interactions significantly outweighed the independent relationship which is assumed by the sum model. This gain of knowledge is one of the major benefits of this combined approach. When interpreting the importance of certain *cis*-regulatory elements, dependencies and interactions between neighbouring *cis*-regulatory elements should be considered. For example, according to our results region A seems to be important for *REN* activation especially in combination with the neighbouring sequences C, D or E. This would mean that if one analyses the role of a native transcription factor targeting region A the other regions need to be considered in this analysis as well.

By tagging the native *REN* with the reporter *firefly luciferase*, the *REN* expression could be analysed in its natural position in the human genome. This is another great advantage over classical reporter studies, as they do not allow investigation in an endogenous context [10–15]. To our knowledge such a combined approach of experimental research of *cis*-regulatory importance and possible statistical interaction by means of a modified Cas9-system and subsequent computational modelling has not been performed before. The only study known to us

that attempted to elucidate the complexity of a *cis*-regulatory element of renin using a model-ling approach has been published in 2007 by Mrowka et. al. However, the endogenous context was not considered in that study [42]. Another advantage of this approach is the induction of gene expression by the used trans-activation complex SAM [34]. This is because the TFs of this complex are always the same. Therefore, any variations in the diversity of transcription result from the *cis*-regions. For instance, if one would use two different transcription factors with two different *cis*-regulatory regions it would not be possible to make a statement about *cis*-regulatory interaction of the region in question.

Of course, each of the generated models have advantages and disadvantages. The sum model is based on fewer parameters than the interaction model. More parameters increase the risk of overfitting of a selected linear model, which is why the sum model has advantages here. When looking at the calculated prediction error, the interaction model shows a smaller value. This fact can also be attributed to the higher number of parameters which the interaction model is based on. Another possibility of comparing the models is the AIC. The AIC is a common criterion for the evaluation of different linear models of the same dataset. On the one hand, it rewards the goodness of fit (likelihood function), but it also contains a penalty term, which penalises too high model complexity. This corresponds to an evaluation based on the principle of maximum parsimony. With an increase in model complexity, the goodness of the fit usually gains as well (risk of overfitting). But the calculated AIC was also smaller for the interaction model. Another big advantage of this approach is that the entire statistical evaluation in the form of model fitting and model evaluation by 2-group Mann-Whitney U Test of the respective prediction errors and calculation of the AIC did not need any further parameters. This does not apply to the experimental parameters such as incubation temperature or the determination of the timepoint of approximately 60 hours after transfection for the modelling approach.

In conclusion, we found that the interaction model explains the measured expression levels of *REN* better. Thus, interactions between the individual sequences seem to be necessary to explain the measured activity levels of *REN*. This in turn would mean that interactions between the individual promoter sections are necessary in order to describe the transcription of *REN* via the proximal *renin* promoter. In this study showed that complex interactions occur within the selected *cis* regions of the endogenous proximal promoter of *REN*, which could be relevant for a better understanding of its transcriptional regulation. This novel combined approach of combinatorial transfections and mathematical modelling and the use of a modern Cas9-based trans-activating complex expands the possibility to study the *cis*-regulatory importance of non-coding DNA in an endogenous context. Further, this approach might be potentially useful to examine other genomic regions.

## Supporting information

**S1 Fig. Cassette stably integrated into the genome of the HEK dCas9-SAM_Renin-luciferase cells.** The cassette contains the Firefly luciferase, which was used as a reporter for REN expression. Puromycin was used to select the cells. The homology arms were required for the correct in frame insertion of the cassette. The elements G418-resistance and thymidine-kinase also contained in the cassette were not used in this study.
(TIF)

**S2 Fig. Strength of influence of the investigated promoter regions based on the sum model.** All investigated sequences significantly affect the activation of REN. Considering the sum model, guide A reached the most potent effect. Significances resulted from the

linear regression modelling with the *lm* function of the *stats* package in R. (p-values: $^*<0,05$, $^{***}<0,0005$).
(TIF)

**S1 Table. Luciferase activity for all 32 guide combinations at three different time points.**
(DOCX)

**S1 Script. R script source code.**
(PDF)

**S1 Data.**
(CSV)

# Acknowledgments

We thank Silke Nossmann, Gesine Hauschild, and Nicole Kaiser for technical assistance in cell culture.

# Author Contributions

**Conceptualization:** Steven Kirchner, Stefanie Reuter, Ralf Mrowka.

**Data curation:** Steven Kirchner.

**Formal analysis:** Steven Kirchner, Ralf Mrowka.

**Investigation:** Steven Kirchner.

**Methodology:** Anika Westphal, Ralf Mrowka.

**Resources:** Stefanie Reuter, Anika Westphal, Ralf Mrowka.

**Software:** Ralf Mrowka.

**Supervision:** Stefanie Reuter, Ralf Mrowka.

**Validation:** Steven Kirchner, Ralf Mrowka.

**Visualization:** Steven Kirchner.

**Writing – original draft:** Steven Kirchner.

**Writing – review & editing:** Steven Kirchner, Stefanie Reuter, Anika Westphal, Ralf Mrowka.

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
