## [Decision Letter · Decision Letter 0]

5 Nov 2019

PONE-D-19-26748

Decipher the complexity of cis-regulatory regions by a modified Cas9

PLOS ONE

Dear Mr. Kirchner,

Thank you for submitting your manuscript to PLOS ONE. After careful consideration, we feel that it has merit but does not fully meet PLOS ONE’s publication criteria as it currently stands. Therefore, we invite you to submit a thoroughly revised version of the manuscript that addresses the points raised during the review process.

We would appreciate receiving your revised manuscript by Dec 20 2019 11:59PM. To enhance the reproducibility of your results, we recommend that if applicable you deposit your laboratory protocols in protocols.io, where a protocol can be assigned its own identifier (DOI) such that it can be cited independently in the future. For instructions see: http://journals.plos.org/plosone/s/submission-guidelines#loc-laboratory-protocols

We look forward to receiving your revised manuscript.

Kind regards,

Hans A Kestler

Academic Editor

PLOS ONE

Journal Requirements:

"Grant Support: Funding: Bundesministerium für Bildung und Forschung (BMBF) grant: FKZ

400 01EK1612B to RM.".

 "NO - The funders had no role in study design, data collection and analysis, decision to publish, or preparation of the manuscript.".

[NO].

Reviewers' comments:

Reviewer's Responses to Questions

**Comments to the Author**

1. Is the manuscript technically sound, and do the data support the conclusions?

Reviewer #1: Yes

Reviewer #2: Partly

2. Has the statistical analysis been performed appropriately and rigorously? 

Reviewer #1: No

Reviewer #2: Yes

3. Have the authors made all data underlying the findings in their manuscript fully available?

Reviewer #1: No

Reviewer #2: Yes

4. Is the manuscript presented in an intelligible fashion and written in standard English?

Reviewer #1: Yes

Reviewer #2: Yes

5. Review Comments to the Author

Reviewer #1: The authors describe an interesting new method of directly targeting and cis-activating proximal promotor regions. They present expression level measurements of thus activated renin, and an analysis of the effect of different proximal promotor regions on the expression level of renin, as well as two computational models made from this data.

Major comments:

Introduction:

1. The function and importance of Renin should be described in the Introduction.

2. Please give details about the general processes of trans- and cis-transcriptional regulation. This would make the manuscript more accessible to non-transcription regulation-experts as well.

3. How does the new trans-activating dCas9-system help analyse cis-activation of the proximal promotor? What are classical reporter studies and how are they done? Please go into more detail on this and mention it in the manuscript.

4. What is RAAS and how does it relate to the experiments described in this manuscript? It is only mentioned once in the introduction without explanation (or an explanation of the abbreviation), and then never mentioned again in the paper.

Materials and Methods:

5. If I understand correctly, Renin was tagged with a sequence for Geniticin resistance, but cell selection was done with Puromycin. How does this work? This needs to be clarified.

6. The models need more explanation and description: Why were these models chosen? How were the models built? Which data was the basis for the models and which data points were used as controls? How were the models analysed? What is the accuracy of your models and how did you compute this value? Is there code available regarding these analyses? How did you do your statistics? All of these questions needs to be included and answered in the Materials and Methods Section.

Discussion:

7. It is shown in your results that guide A has especially good activating ability for Renin. Is the activation level of Renin by guide A comparable to biological activation of Renin? Can you be sure that the activation ability of guide A is representative for all the other guides? Whithout knowing this basing the concentration of all other guides on guide A seems somehow random to me. What was your reasoning for this? Why not base the concentrations each of the guides on a baseline for each of them? Please answer these questions in the manuscript.

8. Is there independent data corroborating the results of the analyses of the models? What are the pros and cons regarding your choice of model. Please discuss this in the manuscript.

9. If I understand correctly, each of the guides targets a certain known area of the proximal promotor. Can you formulate hypotheses on how your data and models relate back to these promotor regions and how they interact to lead to the relative expression levels? Are the regions of guides A and B already known fo being the most important activators of Renin expression or is this a new discovery? Actually, to me most of the Discussion section seems to belong in the results section as it just describes the results but does not discuss them in biological or other contexts. Please include a discussion of these questions in the revised manuscript.

Supplement:

10. Throughout the manuscript there are a few results mentioned without showing the data they are based on. All these experiments and results should be provided in a Supplement file. Also, R code for all model generation and all analyses should be provided as well.

General:

11. There are a few grammar, tense and word choice mistakes. Please revise and proofread the manuscript regarding these, especially the Materials and Methods Section. Also, the symbol for micro was not displayed correctly in many places. Many abbreviations throughout the manuscript were either not explained or only explained later in the text - this must be corrected as to the common abbreviation customs. Also, please provide the full names with locations for each of your Materials suppliers and for all substances used and mentioned in the manuscript.

Minor comments:

12. Are there variations in the transfection performance when using 7ng of DNA compared to 150ng?

13. Figure 2: Why does the luciferase activity decrease at higher guide A concentrations? Please mention this in the Figure text.

14. Figure 3: To me, this Figure seemed redundant in many parts. I through IV are never explained in the figure text, the different combinations were already represented in Table 1, and the actual results are represented in Figure 4. I propose the incorporation of parts of Figure 3 into Figure 4 together with its Figure text. Or reduce Figure 3 into its essential parts.

15. Figure 4: The slight differences in color at time point 60.52h are nearly not visible. Please include numeric values for each combination.

Reviewer #2: The aim of this project is to understand the activity of cis-regulated promoters, which is investigated here using the renin promoter as example. For this purpose, different guides have been developed resulting in binding of the SAM complex to the renin promoter. Consequently, renin expression as well as the expression of the attached firefly luciferase is induced. In order to better understand the interaction of the components involved in the cis-region, two types of mathematical models have been applied and compared to experimental data obtained from experiments.

However, when reading it is not clear to me whether the primary goal was to gain knowledge about the renin promoter or to investigate a new model approach. The presented results seem interesting to me but should be described in much more detail thus reducing the potential for speculations. In addition, it should be clarified at the end which significance the knowledge gained in this project has for the cis-regulator region of the renin promoter.

Major comments:

-Why did you study the renin promoter? Is it a prominent example of cis-regulation or is there a relationship to a disease?

-Laboratory experiments are detailed described, whereas the model setups of the sum model and the interaction model are not described at all. Please go into detail in the methods about the model setup and how simulations were performed based on these models. Furthermore, describe the performed statistical analyses.

In addition, introduce both kinds of models in the introduction. What are differences and limitations? How do they work? In which context are the models otherwise applied?

-The results should be described in much greater detail and initial interpretations should be given. Unfortunately, the explanation of the experiments is described in more detail in the discussion than in the actual part of the results. The figures should also not be assigned a self-explanatory role that allows each reader his own interpretation.

-In figure 2 you show the molecular ratio of guide A to achieve maximal luciferase activity. Do these concentrations levels also correspond to luciferase activity after transfection with the other guides B-E? If not, what is the explanation?

-Figure 5 compares the model predicted values in comparison to measured data. But it is not stated which of the two model is compared. Are there differences between the models? Same question for Figure 6. In addition, the statements in Figures 5 and 6 are almost identical. It should be thought to use only one of the two figures and to present the measured data in a supplement material. Then I would recommend preparing the colored surface depicting luciferase activity for both kinds of model in comparison to real measured data.

- In the discussion, the results obtained should be discussed and not described again.

What biological conclusions can be drawn from the experiments and is this assumption supported by other studies? Do the new findings reflect the state of the art? Which of the two applied models is the best to study promoter activity and why? What came out of the studies regarding renin promoter activity and how can this be used?

Minor comments:

-Put only the real manuscript title on the title page and not the short one necessary for PLOS ONE.

-Add the symbol asterisk for the corresponding author.

-Please refer to the guidelines of PLOS ONE concerning double spaced text, figure references in the text, figure legends and use squared brackets to refer to other publications. See therefore:

https://journals.plos.org/plosone/s/submission-guidelines

https://journals.plos.org/plosone/s/file?id=80c1/PLOSOne_formatting_sample_main_body.pdf

-English should be improved. Try to use shorter sentences and avoid to many “which’s”. Do not mix British and American English and replace the decimal place comma by a point.

-Did the online design tool “crispr.mit.edu” suggest you more than the shown five guides A’-E’ and if so, why did you chose these five guides?

-Figures should be referenced in the sentence and not after the point.

-Fig. 3 is another representation of table 1. Where is the benefit of this figure?

-Fig. 6: Here I would recommend considering the choice of color in order to take color blindness into account.

-Since you study gene expression of renin, I would recommend using the human gene symbol REN for renin instead of the protein name.

6. PLOS authors have the option to publish the peer review history of their article (what does this mean?). If published, this will include your full peer review and any attached files.

Reviewer #1: No

Reviewer #2: Yes: Silke Daniela Kühlwein

---

## [Author Response · Author response to Decision Letter 0]

6 Jan 2020

Dear Editor,

thank you for reviewing the submitted article and for the opportunity to submit a revised version of the manuscript. We have answered all the questions that you and the reviewers have raised. In particular we have extended the introduction section and reorganized the results and discussion in order to make it more accessible to a wider range of readers. Further we have improved the graphics and statistical presentation. We have further given more detailed information about the modelling approach and have provided all the R-scripts in the supplement. 

All authors declare that we do not have competing interests.

Journal Requirements

1. Please ensure that your manuscript meets PLOS ONE's style requirements, phrase “data not shown”

According to the journal requirements we have provided all the data such that the phrase “data not shown” was removed completely. 

2. funding information in the Acknowledgments section 

We removed the funding information in the Acknowledgments section. You will find the funding information in the Funding Statement section of the online submission form. 

3. Competing interests

We added the statement “The authors have declared that no competing interests exist.” in a Competing interests section in our cover letter. 

Review 

Comments to the Author

Reviewer #1: 

Dear reviewer, 

thank you for reviewing the submitted manuscript. We have thoroughly revised the manuscript according to the points that you have raised and have answered all your questions. 

Major comments

Introduction

1. The function and importance of Renin should be described in the Introduction.

We introduced a paragraph with explanations of renin and the renin-angiotensin-aldosterone system (RAAS) and its importance for the human organism. This was done in the introduction section.

2. Please give details about the general processes of trans- and cis-transcriptional regulation. 

We added a paragraph to give details about general processes of trans- and cis-transcriptional regulation. This can be found in the introduction section.

3. How does the new trans-activating dCas9-system help analyse cis-activation of the proximal promotor? 

We have added the following paragraph that emphasizes the importance of the trans-activating complex:

“Apart from gene editing the guided binding of the Cas9-RNA-DNA complex may be used for other purposes than gene editing. This includes gene activation, gene silencing and gene labeling with fluorescent proteins for example. In this work we used a modified Cas9 system that can be used to activate genes. This makes use of non-cutting Cas9 that was fused to an activation transcription factor. We used this system in order to explore the cis-regulatory importance of genomic DNA. More specifically we use the synergistc activation mediator complex (SAM)…

…The development of this new gene-editing tool enables the study of genes and their regulatory elements in the endogenous context and in the natural environment of the gene. This form of investigation is not possible by classical reporter studies (as described above). In addition, the defined and constant trans-factor complex allows the isolated investigation of the cis-regulatory regions, since the same trans-factors always bind to the regulatory elements.”

What are classical reporter studies and how are they done? 

We added the following paragraph to give details about that in the introduction:

“Numerous classical reporter studies have enabled the identification of important cis-regulatory elements. In these studies, restriction pieces of investigated DNA, e.g. promoter regions, are cloned into vectors. After transfection of those vectors into cells, activation levels of these promoter regions can be measured via downstream reporter genes, such as green fluorescent protein (GFP) or luciferase. But the investigated DNA, e.g. the promoter regions are removed from their endogenous context. Thus, the experiments must be performed outside of the natural environment of the promoter. This circumstance and the lack of opportunity to study complex interactions of individual regulatory elements are limitations of these classic reporter studies.”

4. What is RAAS and how does it relate to the experiments described in this manuscript? 

Thank you for raising this issue. This point was introduced in an extra paragraph in the introduction section.

Materials and Methods:

5. If I understand correctly, Renin was tagged with a sequence for Geniticin resistance, but cell selection was done with Puromycin. How does this work? 

The genome was modified in the following way:

Renin-P2A-luciferase-T2A-G418-PGK promoter-puromycin resistance 

In order to select for successful insertion of the insert puromycin may be used. The G418 was introduced to have a potential additional selection marker for selecting renin positive cells in a possible high throughput screening. This G418 resistance cassette was not used in this study.

Therefore, the cloning of G418 resistance has no significance for this work but was cloned for potential further experiments with those cells or plasmids. 

The actual selection was performed with puromycin. 

6. The models need more explanation and description: Why were these models chosen? 

Independent linear models are reasonable way to describe multifactor influences on output variable (in this case renin transcription). The next level of complexity would include apart from the linear scaling factors, those factors that describe statistical interaction. 

How were the models built? 

These models have been fitted with the measured data. For the fittings of both linear models the function lm of the package stats in the R version 1.1.423 was used. 

Which data was the basis for the models and which data points were used as controls? 

The complete data set that we have used is included in the supplement material. Random controls were generated by a section in the R script.

How were the models analyzed? What is the accuracy of your models and how did you compute this value? Is there code available regarding these analyses? How did you do your statistics? 

The lm function of the R package stats computes the linear factors that fit the input variables according to the proposed model including all the statistics of the fitted parameters. The full code of the R scripts is provided in the supplement material. 

Discussion:

7. It is shown in your results that guide A has especially good activating ability for Renin. Is the activation level of Renin by guide A comparable to biological activation of Renin? Can you be sure that the activation ability of guide A is representative for all the other guides? Without knowing this basing the concentration of all other guides on guide A seems somehow random to me. What was your reasoning for this? Why not base the concentrations each of the guides on a baseline for each of them? Please answer these questions in the manuscript.

This experiment was only performed to find a sensible dosage for transfection. In order to make this clear we have added this information to the manuscript. 

Performing all subsequent experiments with multiple concentrations of the individual guides would have allowed the dimensions of the experiments to grow exponentially. With a repetition factor of 6 which was used in our approach, we would have had to perform 18,750 instead of 192 experiments assuming 4 different concentrations of each guide. 

It would have been perhaps interesting to see those results of this extended approach. However, at that scale it was out of our resources. 

8. Is there independent data corroborating the results of the analyses of the models? 

This particular region with this specific setup has not been investigated by other groups (to our knowledge). To put this on a more general level it is of interest of studying gene regulation whether or not particular regulatory elements act independently or whether or not those elements show interactions with other regulatory regions.

What are the pros and cons regarding your choice of model? Please discuss this in the manuscript.

sum model: pro: less parameters contra: larger prediction error 

interaction model: pro: smaller prediction error contra: more parameters

In order to find the optimum according to the principle of maximal parsimony we used the Akaike Information Criterion (AIC) for judging which model to prefer. 

9. If I understand correctly, each of the guides targets a certain known area of the proximal promotor. Can you formulate hypotheses on how your data and models relate back to these promotor regions and how they interact to lead to the relative expression levels? Are the regions of guides A and B already known to being the most important activators of Renin expression or is this a new discovery? 

Thank you for raising this point. As you have suggested we have checked whether or not there are transcription factor binding sites for A and B present or known in the literature. In fact, for region A the so-called cAMP responsive cis element is present which has been investigated to be important for cAMP dependent activation of renin. This fact has been put to the introduction section. 

Actually, to me most of the Discussion section seems to belong in the results section as it just describes the results but does not discuss them in biological or other contexts. Please include a discussion of these questions in the revised manuscript.

Thank you for putting up this issue. We have completely revised the discussion section and have shifted all parts that are related to the results to the results section. 

Supplement:

10. Throughout the manuscript there are a few results mentioned without showing the data they are based on. All these experiments and results should be provided in a Supplement file. Also, R code for all model generation and all analyses should be provided as well.

We provide now the following additional data in the supplement section:

- S1 Fig. Cassette stably integrated into the genome of the HEK dCas9-SAM_Renin-luciferase cells.

- S2 R-script. “source code”

- S3 Fig. Strength of influence of the guides based on the sum model.

General:

11. There are a few grammar, tense and word choice mistakes. Please revise and proofread the manuscript regarding these, especially the Materials and Methods Section. Also, the symbol for micro was not displayed correctly in many places. Many abbreviations throughout the manuscript were either not explained or only explained later in the text - this must be corrected as to the common abbreviation customs. Also, please provide the full names with locations for each of your Materials suppliers and for all substances used and mentioned in the manuscript.

The manuscript has been read by a native speaker in order to correct and avoid possible mistakes.

Minor comments:

12. Are there variations in the transfection performance when using 7ng of DNA compared to 150ng?

The variations are shown in the error bars of the experiments in Fig 2 which represents standard deviation. The dose response curve was used to find the concentration for a certain activation of renin. 

13. Figure 2: Why does the luciferase activity decrease at higher guide A concentrations? Please mention this in the Figure text.

We mentioned the decrease of luciferase activity in the figure text. The increase in the concentration of the guide DNA above 30 ng per well did not lead to a significant increase in expression, presumably due to saturation phenomenon.

14. Figure 3: To me, this Figure seemed redundant in many parts. I through IV are never explained in the figure text, the different combinations were already represented in Table 1, and the actual results are represented in Figure 4. I propose the incorporation of parts of Figure 3 into Figure 4 together with its Figure text. Or reduce Figure 3 into its essential parts.

Thank you for raising this point. To follow your suggestion we have reduced the number of examples in Fig 3. The new Fig contains now less instances of combinations. We also changed the color scheme of the figures. This also should address red-green color blindness.

15. Figure 4: The slight differences in color at time point 60.52h are nearly not visible. Please include numeric values for each combination.

Since the values for expression levels were widely divergent, only a few surfaces would have been colored in a linear scale, whereas most would have remained white. That’s why we chose a logarithmic scale for figure 4. We changed the color scheme of the figure in order to better recognize the differences. We did not use a figure with numerical values, as this would have become very confusing. 

 

Reviewer #2: 

Dear reviewer #2, 

thank you for reviewing the submitted manuscript. We have thoroughly revised the manuscript according to the points that you have raised and have answered all your questions. First, we want to show a novel approach to investigate cis-regulatory interactions. 

Major comments:

• Why did you study the renin promoter? Is it a prominent example of cis-regulation or is there a relationship to a disease? 

Renin is the rate limiting step in the renin-angiotensin-aldosterone-system (RAAS), which is important for the long-term regulation of blood pressure. According to WHO high blood pressure for example is one of the major causes of premature death worldwide (https://www.who.int/news-room/fact-sheets/detail/hypertension).

Hence understanding of transcriptional regulation is potentially important for understanding basic principles of many cardio-vascular diseases. These points are introduced in a paragraph with explanations of renin and the renin-angiotensin-aldosterone system (RAAS) and its importance for the human organism in the introduction section.

• Laboratory experiments are detailed described, whereas the model setups of the sum model and the interaction model are not described at all. Please go into detail in the methods about the model setup and how simulations were performed based on these models. 

Furthermore, describe the performed statistical analyses. 

Both models are introduced int the introduction section. For details of the implementation including the statistical analysis we provide the complete R-script in the supplement. The basic ideas of the models including their pros and cons have now been added to the manuscript. 

In addition, introduce both kinds of models in the introduction. 

Thank you for raising this point. We added a paragraph to the introduction section that describes both models.

What are differences and limitations? How do they work? 

We added the following paragraph to the discussion section:

“Independent linear models are a way to describe multifactor influences on output variable (in this case renin transcription) …

… The next level of complexity would include apart from the linear scaling factors, those factors that describe statistical interaction. Therefore, an interaction model was subsequently computed and fitted to the measured data. The selected models each have advantages and disadvantages. The sum model is based on fewer parameters than the interaction model. More parameters increase the risk of overfitting of a selected linear model, which is why the sum model has advantages here. When looking at the calculated prediction error, the interaction model showed a smaller value. This fact can also be attributed to the higher number of parameters which the interaction model is possibly based on.”

• In which context are the models otherwise applied?

An example of a linear sum model would be the calculation of the melting temperature of the DNA:

Tm = 4(G+C) +2(A+T)

And in a more advanced complex form a formula that considers the neighboring relationships of the basepairs:

Hooyberghs, J.; Van Hummelen, P.; Carlon, E. (2009). "The effects of 

mismatches on hybridization in DNA microarrays: Determination of nearest 

neighbor parameters". Nucleic Acids Research. 37 (7): e53. 

doi:10.1093/nar/gkp109. PMC 2673445. PMID 19270064

In general the question of independence or interaction is an important question of many multifactor analysis setups in science [1]. 

In the simplest case this would be a two-factor setup, for example whether or not a drug (factor 1) acts dependent on sex (factor 2). 

• The results should be described in much greater detail and initial interpretations should be given. Unfortunately, the explanation of the experiments is described in more detail in the discussion than in the actual part of the results.

In order to give more details about the results we have restructured the results and discussion section. Furthermore, we have given more details in Fig 5 describing the original data and the prediction values of both models.

We have modified the captions of the figures in order to enhance the readability. 

• In figure 2 you show the molecular ratio of guide A to achieve maximal luciferase activity. Do these concentrations levels also correspond to luciferase activity after transfection with the other guides B-E? If not, what is the explanation?

This dose-response experiment should only determine the concentration under which a certain activation of renin occurs. Performing all subsequent experiments with multiple concentrations of the individual guides would have allowed the dimensions of the experiments to grow exponentially. With a repetition factor of 6 which was used in our approach, we would have had to perform 18,750 instead of 192 experiments assuming 4 different concentrations of each guide. 

It would have been perhaps interesting to see those results of this extended approach. However, at that scale it was out of our resources. 

• Figure 5 compares the model predicted values in comparison to measured data. But it is not stated which of the two model is compared. 

Are there differences between the models? Same question for Figure 6. In addition, the statements in Figures 5 and 6 are almost identical. It should be thought to use only one of the two figures and to present the measured data in a supplement material. 

In order to provide better understanding of the models and the data we provide first a time dependent Fig 5 and a cross sectional representation of 3 selected timepoints highlighting the overlapping values in a venn diagram representation. We believe that both figures enhance the understanding of the data and the modeling approach. 

Then I would recommend preparing the colored surface depicting luciferase activity for both kinds of model in comparison to real measured data.

Thank you for highlighting this point. We have implemented now your suggestion in Fig 5 and 6. 

• In the discussion, the results obtained should be discussed and not described again.

Thank you for raising this point. We have revised the results and discussion section and separated results from discussion more clearly. 

• What biological conclusions can be drawn from the experiments and is this assumption supported by other studies? Do the new findings reflect the state of the art? 

Understanding of the cis-regulatory role is of general interest. There are examples were mutations of cis-regulatory promoter regions lead to sever diseases [2]. Many studies of promoter activity have been done by means of isolated singular promoter fragments that have been cloned in reporter vectors. In our approach we investigate the promoter region in its natural context. This was done by the recently developed modified Cas9/SAM (synergistic activation mediator) complex. We further expanded this to a combinatorial approach. The advantage of using the SAM system is that the trans factors are always the same. Therefore, any variations in the transcription variety must be due to the cis region. If one would for instance use two different transcription factors with two different cis regulatory regions it would not be possible to make a statement about cis-regulatory interaction.

• Which of the two applied models is the best to study promoter activity and why? 

Our analysis shows that the interaction model is best to describe the experimental data obtained in our study. First the prediction error is lower and second because we want to obey the principle of maximum parsimony, we have checked the models with the Akaike information criterion (AIC). The AIC is an estimator of out-of-sample prediction error and thereby relative quality of statistical models for a given set of data. Given a collection of models for the data, AIC estimates the quality of each model, relative to each of the other models. Thus, AIC provides a means for model selection.

• What came out of the studies regarding renin promoter activity and how can this be used?

Thank you for raising this issue. For example, region A is important especially in combination with neighboring regions, this would mean if one investigates role of a native transcription factor targeting region A the other regions must be considered in this analysis as well. To make this clearer we have added this interpretation to the discussion. 

Minor comments:

• Put only the real manuscript title on the title page and not the short one necessary for PLOS ONE.

We removed the short title from the title page.

• Add the symbol asterisk for the corresponding author.

We added the symbol asterisk for the corresponding author.

• Please refer to the guidelines of PLOS ONE concerning double spaced text, figure references in the text, figure legends and use squared brackets to refer to other publications. See therefore:

Thank you for raising this point. We have revised the manuscript in order to fulfill the guidelines of PLOS ONE.

• English should be improved. Try to use shorter sentences and avoid to many “which’s”. Do not mix British and American English and replace the decimal place comma by a point.

The manuscript has been read by a native speaker in order to correct and avoid possible mistakes.

• Did the online design tool “crispr.mit.edu” suggest you more than the shown five guides A’-E’ and if so, why did you chose these five guides?

We have focused the study on the proximal promoter of renin comprising approximately the first 200 bases from transcription start site. In this region we have chosen 5 target regions appr. evenly distributed with the help of the “crispr.mit.edu” online tool. Detailed information about sequences and their positions is given in the manuscript.

• Figures should be referenced in the sentence and not after the point.

We avoided to refer figures after the point in the revised manuscript.

• Fig. 3 is another representation of table 1. Where is the benefit of this figure?

Fig. 3 is intended to be a better illustration of the 32 possible combinations and serves to better understand the following Figures 4 and 6.

• Fig. 6: Here I would recommend considering the choice of color in order to take color blindness into account.

We changed the colors of Fig. 3, 4 and 6. This also should address red-green color blindness.

• Since you study gene expression of renin, I would recommend using the human gene symbol REN for renin instead of the protein name.

We replaced “renin” or “human renin” with REN in the manuscript.

 

1. Nieuwenhuis S, Forstmann BU, Wagenmakers EJ. Erroneous analyses of interactions in neuroscience: a problem of significance. Nat Neurosci. 2011;14(9):1105-7.

2. Soufi M, Ruppert V, Rinné S, Mueller T, Kurt B, Pilz G, et al. Increased KCNJ18 promoter activity as a mechanism in atypical normokalemic periodic paralysis. Neurol Genet. 2018;4(5):e274-e.

---

## [Decision Letter · Decision Letter 1]

20 Feb 2020

PONE-D-19-26748R1

Decipher the complexity of cis-regulatory regions by a modified Cas9

PLOS ONE

Dear Mr. Kirchner,

Thank you for submitting your manuscript to PLOS ONE. After careful consideration, we feel that it has merit but does not fully meet PLOS ONE’s publication criteria as it currently stands. Therefore, we invite you to submit a revised version of the manuscript that addresses the points raised during the review process.

I think the reviewers have made their points quite clear, please address these issues.

We would appreciate receiving your revised manuscript by Apr 05 2020 11:59PM. To enhance the reproducibility of your results, we recommend that if applicable you deposit your laboratory protocols in protocols.io, where a protocol can be assigned its own identifier (DOI) such that it can be cited independently in the future. For instructions see: http://journals.plos.org/plosone/s/submission-guidelines#loc-laboratory-protocols

We look forward to receiving your revised manuscript.

Kind regards,

Hans A Kestler

Academic Editor

PLOS ONE

Reviewers' comments:

Reviewer's Responses to Questions

**Comments to the Author**

1. If the authors have adequately addressed your comments raised in a previous round of review and you feel that this manuscript is now acceptable for publication, you may indicate that here to bypass the “Comments to the Author” section, enter your conflict of interest statement in the “Confidential to Editor” section, and submit your "Accept" recommendation.

Reviewer #1: (No Response)

Reviewer #2: (No Response)

2. Is the manuscript technically sound, and do the data support the conclusions?

Reviewer #1: Partly

Reviewer #2: No

3. Has the statistical analysis been performed appropriately and rigorously? 

Reviewer #1: I Don't Know

Reviewer #2: Yes

4. Have the authors made all data underlying the findings in their manuscript fully available?

Reviewer #1: Yes

Reviewer #2: Yes

5. Is the manuscript presented in an intelligible fashion and written in standard English?

Reviewer #1: (No Response)

Reviewer #2: Yes

6. Review Comments to the Author

Reviewer #1: I thank the authors for reworking their manuscript in accordance with the comments made in my previous review! Most of my suggestions and comments have been included, though some issues remain.

Major comments:

1. Trans-regulatory elements and their function are not described, only mentioned in context of experimental setup. Please describe their basic function and occurrance in the introduction.

2. Providing the R-Source-Code in the Supplementary Materials is not a valid substitution for the description of model choice, the modelling processes and validation, and the experiments done on then. Without a detailed description of the models (etc.) no clear evaluation of the data in your manuscript can be made.

- Please provide sources stating that independent models are indeed a valid way to describe multifactor variables on an output factor, and describe why you decided on this model type.

- Instead of mentioning the R functions used in the script, describe how the fitting was done, how the random samples were generated, which data was used for the models in which way, how the models were evaluated (e.g. the error calculations mentioned in the manuscript), which statistical tests were used and with what parameters and why, and how the experiments were performed.

- Cite R and the packages used.

- Describe all of the above in the manuscript in a manner as to be reproducible by potential readers, without them having to use the provided R-Script.

3. Regarding the previous major comment 7 of mine, there are still some questions which are left unanswered. Please answer them and include the answers in the manuscript:

Is the activation level of Renin by guide A comparable to biological activation of Renin? Can you be sure that the activation ability of guide A is representative for all the other guides? Without knowing this basing the concentration of all other guides on guide A seems somehow random to me. What was your reasoning for this?

4. Please provide citations for the hypotheses and assumptions in the discussion section.

Minor comments:

5. In the Manuscript, there are instances where references were not found and are included in the text in the form of "Error! Reference source not found.". Please revise the sourcing in your manuscript.

6. Some of the figure descriptions do not fit the new figures anymore.

7. American and British English are still mixed. Please review the manuscript again regarding this.

Reviewer #2: The manuscript has improved. However, not all the comments made previously have been adequately addressed. In general, it is not enough to answer the raised questions within the answer to the reviewer. Please include these previous raised issues in the manuscript.

Especially, I miss in the discussion a comparison of the applied model type to other available models as well as an explanation for the impact of your findings on the RAAS system and related diseases. Please compare your findings with the current knowledge and the outcome of other studies.

The model setup and statistical analyses are still not sufficiently described.

It is not enough to show a R Code that lacks detailed model representations. Based on the given data, it is infeasible to get even a rudimentary understanding of the models. I would expect an overview of the models and a brief explanation about their establishment.

Furthermore, te added part within the introduction does not help to understand the model type at all.

Different statistical tests were applied (e.g. Wilcoxon and ANOVA) without naming them in the method section. If a p-value is given in the results, it should be clear which test was used to calculate it.

I miss references in the discussion that support the assumptions. Likewise, the method section misses references for the applied R packages as well as for R on its own.

In the discussion and not in the method section, I learned that you used the guides with the best score. What was the scoring threshold?

English was improved, but there is still a huge mix of British and American English through the whole manuscript.

Fig.3: color of the plot is not grey and not blue as stated in the figure legend.

7. PLOS authors have the option to publish the peer review history of their article (what does this mean?). If published, this will include your full peer review and any attached files.

Reviewer #1: No

Reviewer #2: No

---

## [Author Response · Author response to Decision Letter 1]

4 Apr 2020

Dear Editor,

thank you for reviewing the revised article and for the opportunity to submit a further revised version of the manuscript. We have answered all the questions the reviewers have raised. Especially, we have revised the introduction section and have now addressed the points raised in more detail. In addition, the discussion was reorganized and the points raised were added. Further, we have given more detail about the modelling approach and the statistical evaluation. 

 

Reviewer #1: 

I thank the authors for reworking their manuscript in accordance with the comments made in my previous review! Most of my suggestions and comments have been included, though some issues remain.

Major comments:

1. Trans-regulatory elements and their function are not described, only mentioned in context of experimental setup. Please describe their basic function and occurrence in the introduction. 

Thank you for raising this point. For more understanding we added the following paragraph to the manuscript:

“… In addition to the cis-regulatory elements, the transcription factors (TFs) are essential to enable gene transcription. These proteins bind to DNA and can activate or repress the transcription of genes. There are differences in the way TFs act to regulate gene expression. Some TFs need to assemble with other proteins, others can directly recruit RNA polymerase which then leads to gene transcription [1]. In a current review, a distinction is made between approx. 1600 human TFs, which represent ~8% of all human genes [2]. There are several ways to divide TFs. In general, a division into basal or general TFs and specific transcription factors is possible. Basal TFs are ubiquitous in all cells and necessary for transcription to occur [3]. By assembling they a part of the preinitiation complex that enables the binding of the RNA polymerase and thus the initiation of the transcription via specific DNA binding sites such as TATA boxes [3, 4]. In contrast, specific transcription factors are active only in specific tissues and/or at specific developmental stages. They may bind at specific DNA binding sites (cis-regulatory regions), e.g. promoters, enhancers or silencers and are necessary for the regulation of central mechanisms such as cell development or the response to stimuli via signal cascades [5, 6]. …” 

 

2. Providing the R-Source-Code in the Supplementary Materials is not a valid substitution for the description of model choice, the modelling processes and validation, and the experiments done on them. Without a detailed description of the models (etc.) no clear evaluation of the data in your manuscript can be made.

Thank you for raising this point. in order to clarify this, we provide now all the steps and the formulas in the text in order to describe the models independent of the source code. (see also below):

Please provide sources stating that independent models are indeed a valid way to describe multifactor variables on an output factor, and describe why you decided on this model type

Thank you for raising this point. We have now included a reference addressing this issue of independence and interaction in biological context in the manuscript [7]. 

Regarding the deeper description of the models we have added for instance the following paragraph in the appropriate sections in the manuscript:

“… When statistically examining factors influencing a variable, it is of general scientific interest to find out whether these factors act independently or interdependently. In our opinion, this question of complex interactions was given too little importance in the research on cis-regulatory regions of REN, especially on the proximal promoter [8, 9]. A study that attempted to address this problem with a modelling approach emerged in 2007 [10]. But even here the experiments were plasmid-based and therefore outside of the endogenous context. In addition, a regulatory region of REN was in the focus, which is approximately 14 kbp upstream of the start of the transcription. 

With this study we want to show on the one hand a novel approach to investigate interactions of cis-regulatory regions in an endogenous context, and on the other hand to raise awareness of how important these interactions are for the interpretation of the importance for the transcriptional regulation. 

To study possible complex interactions within the proximal promoter region of REN we applied a novel combined approach. Firstly, this approach consists of combinatorial transfections from five selected guide-RNAs that translocate the SAM-complex to a specific region of the endogenous proximal promoter. The resulting expression levels of REN through the different combinations of the targeted promoter regions can be quantified by a luciferase activity. Secondly, we generated and fitted two different mathematical models to our experimental data. When modelling experimental data, the principle of simplicity should always be considered. The simplest assumption would be that the regions examined influence the expression of REN completely independently of one another. This is reflected in our first model (sum model). The sum model describes an independent relationship of the promoter sequences we examined to their influence on the activation of REN. In this model each individual region is analyzed in respect to its influence on gene expression. This approach is comparable to the knowledge that classic promoter studies can provide, since, as described above, these do not allow the possibility of studying complex interactions. But, an important scientific question in modelling is as to whether there exists statistical interaction [7]. In order to address this issue, a second mathematical model was generated (interaction model). This model represents a more complex assumption of the conditions in the region of the proximal promoter. The interaction model additionally allows interactions between the selected promoter regions to explain the REN activation. Regarding modelling we used a multiple linear regression model to fit the linear parameters, which is a standard statistical method. In order to check which of the generated models can best explain the measured data, the respective absolute prediction error was calculated. Following the principle of maximum parsimony in modelling, the respective Akaike information criterion (AIC) was calculated for further model judging.“… For the independent sum model, the measured activity y_i can be represented by formula 1:

y_i=∈_i+β_0+A_i β_1+B_i β_2+C_i β_3+D_i β_4+E_i β_5 …“

where β_0 is the offset, β_(1…5) are the linear coefficients, A_i…E_i are {0,1} depending on presence in experiment i and ∈_i is the error. 

“… For the interaction model, the measured activity y_i for all the 32 possible combinations can be represented by formula 2: 

y_i=∈_i+β_0+β_1 A_i+β_2 B_i+β_3 C_i+β_4 D_i+β_5 E_i+β_6 A_i B_i+β_7 A_i C_i+β_8 B_i C_i+β_9 A_i D_i+β_10 B_i D_i+β_11 C_i D_i+β_12 A_i E_i+β_13 B_i E_i+β_14 C_i E_i+β_15 D_i E_i+β_16 A_i B_i C_i+β_17 A_i B_i D_i+β_18 A_i C_i D_i+β_19 B_i C_i D_i+β_20 A_i B_i E_i+β_21 A_i C_i E_i+β_22 B_i C_i E_i+β_23 A_i D_i E_i+β_24 B_i D_i E_i+β_25 C_i D_i E_i+β_26 A_i B_i C_i D_i+β_27 A_i B_i C_i E_i+β_28 A_i B_i D_i E_i+β_29 A_i C_i D_i E_i+β_30 B_i C_i D_i E_i+β_31 A_i B_i C_i D_i E_i

where β_0 is the offset, β_(1…31) are the linear coefficients, A_i…E_i are {0,1} depending on presence in experiment i and ∈_i is the error.

 

Instead of mentioning the R functions used in the script, describe how the fitting was done, how the random samples were generated, which data was used for the models in which way, how the models were evaluated (e.g. the error calculations mentioned in the manuscript), which statistical tests were used and with what parameters and why, and how the experiments were performed

Thank you for raising this point. To clarify this the following text has been added to the manuscript:

“… For fitting of the models represented in formula 1 and 2 the fitting functionality of the lm function of the built-in stats package of R version 1.1.423 was used [11]. The idea behind this is to minimize the error in prediction of y by optimizing the linear factors β for the experimental data. The lm function of the R package stats computes the linear factors β that fit the input variables A to E according to the proposed model including all the statistics of the fitted parameters. The complete data that was used including the R-script can be found in the supplementary material (S2 R-Script). Data was analyzed at 60 hours after transfection. 

For the evaluation of the models we used the obtained coefficients of the respective model and put the coefficients in formula 1 and 2 respectively. For statistical analysis of the coefficients itself the built-in p-value calculation of the multiple linear regression of the lm function of R was used. For comparison of the models we calculated the respective absolute prediction errors of the two models with the built-in predict function of the stats package of R [11]. After fitting of the linear factors β, the error ∈_i could be calculated for the experiments i through conversion of the respective formula. The distribution of the errors of the individual experiments are shown in Fig 7. For statistical analysis of the respective absolute prediction errors we used the independent 2-group Mann-Whitney U Test [11]. To compare and to judge the generated models following the principle of maximum parsimony in modelling, the AIC (Akaike Information Criterion) of each model were calculated for further model judging. The AIC was calculated in the R environment with the AIC function of the stats package [11].

Random controls were generated by the built-in sample function of R [11]. For this purpose, a dataset was created by randomizing the values of the respective variables A to E. This randomized data was then fitted to model 1 and 2 and no statistically significant p values for the coefficients were obtained in the random case. (S2 R-script)

The entire statistical evaluation in the form of model fitting and model evaluation by the 2-group Mann-Whitney U Test of the respective prediction errors and calculation of the AIC did not need any further parameters, apart from the experimental parameters such as incubation temperature or evaluation at about 60 hours after transfection. …”

Cite R and the packages used.

In the new version of the manuscript we have cited R, R packages and the appropriate used functions.

Describe all of the above in the manuscript in a manner as to be reproducible by potential readers, without them having to use the provided R-Script.

Thank you for raising this point. We are confident that with the improved version of the manuscript a reader can now replicate the analysis with the information given above without the R script in the supplement.

3. Regarding the previous major comment 7 of mine, there are still some questions which are left unanswered. Please answer them and include the answers in the manuscript:

Is the activation level of Renin by guide A comparable to biological activation of Renin? 

Thank you for raising this point. We added the following paragraph to the manuscript:

“… In a previous unpublished study, we have determined the sensible range of guide RNA for renin expression after activation of this promoter region. We were able to show that the protein renin was present in the supernatant of the cell culture. This suggests that we operate in a biologically sensible range of renin expression. Further it shows that secreted renin (biologically relevant for blood pressure regulation) was produced in our cells. …”

Can you be sure that the activation ability of guide A is representative for all the other guides? Without knowing this basing the concentration of all other guides on guide A seems somehow random to me. What was your reasoning for this?

Thank you for raising this point. We added the following paragraphs to the appropriate section of the manuscript: 

“…From those previous studies including studies of other genes in our lab we have found that concentrations of approx. 30 ng per well (of a 96 well plate) of guide RNAs showed a biologically relevant expression of RNA and protein. That is why we have decided to use the concentration of 30 ng per well for transfection in our combinatorial setup in this study. …”

In order not to confuse the reader we removed figure 2 from the manuscript. 

The determination of the activation ability of the individual promoter regions can be estimated by the resulting expressing levels of the experiments in which the guides were transfected individually. The different measured values for the individual transfections can depend not only on the possible different activation ability but also on other factors that were not considered in this work. For example, the regions examined can be occupied by other TFs. A special secondary structure of the DNA or the arrangement of the nucleosomes can also make the access to the SAM complex more difficult [12].

4. Please provide citations for the hypotheses and assumptions in the discussion section.

Thank you for raising this point. We revised the discussion section and added citations to support our assumptions, for instance: 

“… 99% decrease in the transcriptional activity of renin was also found when the region of the proximal promoter was deleted in reporter assays performed in As4.1 cells [9]. …”

“… Most of the knowledge about the importance of cis-regulatory areas of the REN is based on this type of promoter studies [8, 13]. …” 

“… whether there are interactions within cis-regulatory regions should be a central question in the research of the complex transcriptional network [7]. …”

“… This is another great advantage over classical reporter studies, as they do not allow investigation in an endogenous context [14-19]. …”

Minor comments:

5. In the Manuscript, there are instances where references were not found and are included in the text in the form of "Error! Reference source not found.". Please revise the sourcing in your manuscript.

Sorry for this technical issue and thank you for raising this point. We thoroughly revised the manuscript in order to avoid such errors.

6. Some of the figure descriptions do not fit the new figures anymore.

Thank you for raising this point. We thoroughly revised the captions of the figures in the new manuscript. 

7. American and British English are still mixed. Please review the manuscript again regarding this.

Thank you for raising this point. The manuscript was again revised by another native speaker in order to improve language and to avoid a mix of American and British English. 

 

Reviewer #2: The manuscript has improved. However, not all the comments made previously have been adequately addressed. In general, it is not enough to answer the raised questions within the answer to the reviewer. Please include these previous raised issues in the manuscript. 

Thank you for raising this point. We thoroughly revised the manuscript and answered all the comments made previously. We also added the answers of the previously raised questions to the new manuscript. We would like to address the following previous issue again at this point:

In figure 2 you show the molecular ratio of guide A to achieve maximal luciferase activity. Do these concentration levels also correspond to luciferase activity after transfection with the other guides B-E? If not, what is the explanation?

Thank you for raising this point. We added the following paragraph to the manuscript:

“… We have not performed a concentration dependent analysis for each individual guide in this study. From other studies of other genes in our lab we have used similar concentrations for transfection of guide RNAs that showed biological relevant expression of RNA and protein. That is why we have used the concentration of 30 ng per well for transfection in the combinatorial setup. …”

In order not to confuse the reader we removed figure 2 from the manuscript. 

The determination of the activation ability of the individual promoter regions can be estimated by the resulting expressing levels of the experiments in which the guides were transfected individually. The different measured values for the individual transfections can depend not only on the possible different activation ability but also on other factors that were not considered in this work. For example, the regions examined can be occupied by other TFs. A special secondary structure of the DNA or the arrangement of the nucleosomes can also make the access to the SAM complex more difficult [12].

 

Especially, I miss in the discussion a comparison of the applied model type to other available models as well as an explanation for the impact of your findings on the RAAS system and related diseases. Please compare your findings with the current knowledge and the outcome of other studies.

Thank you for raising this point. Most of the outcome of other studies about the cis-regulatory regions of human Renin refer to plasmid-based classical promoter assays, which assume an independent relationship of the investigated regions. This is due to their experimental setup. By design of their studies it cannot be determined whether an interaction occurs. Further, these assays must be performed outside of the endogenous context of the region of interest. To our knowledge our study is the first that addresses this question and beyond in a natural genomic context of a promoter. In that regard our study investigates the cis-regulatory importance of the proximal renin promoter in a more natural setting. 

These points can be found now in more detail in the new manuscript.

The model setup and statistical analyses are still not sufficiently described.

It is not enough to show a R Code that lacks detailed model representations. Based on the given data, it is infeasible to get even a rudimentary understanding of the models. I would expect an overview of the models and a brief explanation about their establishment. Furthermore, the added part within the introduction does not help to understand the model type at all. 

Thank you for raising this point. We revised the introduction and material and methods sections and added the following paragraphs to get a better understanding of our approach and the reason for generating the two models:

“… When statistically examining factors influencing a variable, it is of general scientific interest to find out whether these factors act independently or interdependently. In our opinion, this question of complex interactions was given too little importance in the research on cis-regulatory regions of REN, especially on the proximal promoter [8, 9]. A study that attempted to address this problem with a modelling approach emerged in 2007 [10]. But even here the experiments were plasmid-based and therefore outside of the endogenous context. In addition, a regulatory region of REN was in the focus, which is approximately 14 kbp upstream of the start of the transcription. 

With this study we want to show on the one hand a novel approach to investigate interactions of cis-regulatory regions in an endogenous context, and on the other hand to raise awareness of how important these interactions are for the interpretation of the importance for the transcriptional regulation. 

To study possible complex interactions within the proximal promoter region of REN we applied a novel combined approach. Firstly, this approach consists of combinatorial transfections from five selected guide-RNAs that translocate the SAM-complex to a specific region of the endogenous proximal promoter. The resulting expression levels of REN through the different combinations of the targeted promoter regions can be quantified by a luciferase activity. Secondly, we generated and fitted two different mathematical models to our experimental data. When modelling experimental data, the principle of simplicity should always be considered. The simplest assumption would be that the regions examined influence the expression of REN completely independently of one another. This is reflected in our first model (sum model). The sum model describes an independent relationship of the promoter sequences we examined to their influence on the activation of REN. In this model each individual region is analyzed in respect to its influence on gene expression. This approach is comparable to the knowledge that classic promoter studies can provide, since, as described above, these do not allow the possibility of studying complex interactions. But, an important scientific question in modelling is as to whether there exists statistical interaction [7]. In order to address this issue, a second mathematical model was generated (interaction model). This model represents a more complex assumption of the conditions in the region of the proximal promoter. The interaction model additionally allows interactions between the selected promoter regions to explain the REN activation. Regarding modelling we used a multiple linear regression model to fit the linear parameters, which is a standard statistical method. In order to check which of the generated models can best explain the measured data, the respective absolute prediction error was calculated. Following the principle of maximum parsimony in modelling, the respective Akaike information criterion (AIC) was calculated for further model judging. …”

“… For the independent sum model, the measured activity y_i can be represented by formula 1:

y_i=∈_i+β_0+A_i β_1+B_i β_2+C_i β_3+D_i β_4+E_i β_5

where β_0 is the offset, β_(1…5) are the linear coefficients, A_i…E_i are {0,1} depending on presence in experiment i and ∈_i is the error. …” 

For the interaction model the measured activity y_i for all the 32 possible combinations can be represented by formula 2: 

y_i=∈_i+β_0+β_1 A_i+β_2 B_i+β_3 C_i+β_4 D_i+β_5 E_i+β_6 A_i B_i+β_7 A_i C_i+β_8 B_i C_i+β_9 A_i D_i+β_10 B_i D_i+β_11 C_i D_i+β_12 A_i E_i+β_13 B_i E_i+β_14 C_i E_i+β_15 D_i E_i+β_16 A_i B_i C_i+β_17 A_i B_i D_i+β_18 A_i C_i D_i+β_19 B_i C_i D_i+β_20 A_i B_i E_i+β_21 A_i C_i E_i+β_22 B_i C_i E_i+β_23 A_i D_i E_i+β_24 B_i D_i E_i+β_25 C_i D_i E_i+β_26 A_i B_i C_i D_i+β_27 A_i B_i C_i E_i+β_28 A_i B_i D_i E_i+β_29 A_i C_i D_i E_i+β_30 B_i C_i D_i E_i+β_31 A_i B_i C_i D_i E_i

where β_0 is the offset, β_(1…31) are the linear coefficients, A_i…E_i are {0,1} depending on presence in experiment i and ∈_i is the error.

Different statistical tests were applied (e.g. Wilcoxon and ANOVA) without naming them in the method section. If a p-value is given in the results, it should be clear which test was used to calculate it. 

Thank you for raising this point. We revised the paragraph “Modelling of received data and statistics” in order to get a better understanding of the generated models and where all the statistics are described. 

I miss references in the discussion that support the assumptions. 

Thank you for raising this point. We revised the discussion section and added citations to support our assumptions, for instance:

“… 99% decrease in the transcriptional activity of renin was also found when the region of the proximal promoter was deleted in reporter assays performed in As4.1 cells [9]. …”

“… Most of the knowledge about the importance of cis-regulatory areas of the REN is based on this type of promoter studies [8, 13]. …” 

“… whether there are interactions within cis-regulatory regions should be a central question in the research of the complex transcriptional network [7]. …”

“… This is another great advantage over classical reporter studies, as they do not allow investigation in an endogenous context [14-19]. …”

 

Likewise, the method section misses references for the applied R packages as well as for R on its own.

All the R packages and R functions have been cited in full in the new manuscript. 

In the discussion and not in the method section, I learned that you used the guides with the best score. What was the scoring threshold?

We have used the “SAM sgRNA design tool”: 

http://sam.genome-engineering.org/database/

The guides are sorted in order of specificity (highest to lowest) based on method described in Hsu et. al. Nature Biotech 2014 [20]. We have chosen to use the top 5 hits for human REN.

We revised the paragraph “REN-guides: Design and Cloning” in the material and methods section in order to make the design understandable.

English was improved, but there is still a huge mix of British and American English through the whole manuscript. 

Thank you for raising this point. The manuscript was again revised by another native speaker in order to improve language and to avoid a mix of American and British English. 

Fig.3: color of the plot is not grey and not blue as stated in the figure legend.

Thank you for raising this point. We revised the figure captions.

 

1. Frietze S, Farnham PJ. Transcription factor effector domains. Subcell Biochem. 2011;52:261-77.

2. Lambert SA, Jolma A, Campitelli LF, Das PK, Yin YM, Albu M, et al. The Human Transcription Factors. Cell. 2018;172(4):650-65.

3. Reese JC. Basal transcription factors. Curr Opin Genet Dev. 2003;13(2):114-8.

4. Shilatifard A, Conaway RC, Conaway JW. The RNA polymerase II elongation complex. Annu Rev Biochem. 2003;72:693-715.

5. Lobe CG. TRANSCRIPTION FACTORS AND MAMMALIAN DEVELOPMENT. Curr Top Dev Biol. 1992;27:351-83.

6. Pawson T. SIGNAL-TRANSDUCTION - A CONSERVED PATHWAY FROM THE MEMBRANE TO THE NUCLEUS. Dev Genet. 1993;14(5):333-8.

7. Nieuwenhuis S, Forstmann BU, Wagenmakers EJ. Erroneous analyses of interactions in neuroscience: a problem of significance. Nat Neurosci. 2011;14(9):1105-7.

8. Glenn ST, Jones CA, Gross KW, Pan L. Control of rene gene expression. Pflugers Arch. 2013;465(1):13-21.

9. Pan L, Gross KW. Transcriptional regulation of renin - An update. Hypertension. 2005;45(1):3-8.

10. Mrowka R, Steege A, Kaps C, Herzel H, Thiele BJ, Persson PB, et al. Dissecting the action of an evolutionary conserved non-coding region on renin promoter activity. Nucleic Acids Res. 2007;35(15):5120-9.

11. Team RC. R: A Language and Environment for Statistical Computing. R Foundation for Statistical Computing; 2017.

12. Venkatesh S, Workman JL. Histone exchange, chromatin structure and the regulation of transcription. Nat Rev Mol Cell Bio. 2015;16(3):178-89.

13. Castrop H, Hocherl K, Kurtz A, Schweda F, Todorov V, Wagner C. Physiology of Kidney Renin. Physiol Rev. 2010;90(2):607-73.

14. Borensztein P, Germain S, Fuchs S, Philippe J, Corvol P, Pinet F. CIS-REGULATORY ELEMENTS AND TRANS-ACTING FACTORS DIRECTING BASAL AND CAMP-STIMULATED HUMAN RENIN GENE-EXPRESSION IN CHORIONIC CELLS. CircRes. 1994;74(5):764-73.

15. Petrovic N, Black TA, Fabian JR, Kane C, Jones CA, Loudon JA, et al. Role of proximal promoter elements in regulation of renin gene transcription. J Biol Chem. 1996;271(37):22499-505.

16. Voigtlander T, Ganten D, Bader M. Transcriptional regulation of the rat renin gene by regulatory elements in intron I. Hypertension. 1999;33(1):303-11.

17. Shi Q, Black TA, Gross KW, Sigmund CD. Species-specific differences in positive and negative regulatory elements in the renin gene enhancer. CircRes. 1999;85(6):479-88.

18. Pan L, Black TA, Shi Q, Jones CA, Petrovic N, Loudon J, et al. Critical roles of a cyclic AMP responsive element and an E-box in regulation of mouse renin gene expression. J Biol Chem. 2001;276(49):45530-8.

19. Pan L, Jones CA, Glenn ST, Gross KW. Identification of a novel region in the proximal promoter of the mouse renin gene critical for expression. Am J Physiol-Renal Physiol. 2004;286(6):F1107-F15.

20. Hsu PD, Lander ES, Zhang F. Development and Applications of CRISPR-Cas9 for Genome Engineering. Cell. 2014;157(6):1262-78.

---

## [Decision Letter · Decision Letter 2]

22 Apr 2020

PONE-D-19-26748R2

Decipher the complexity of cis-regulatory regions by a modified Cas9

PLOS ONE

Dear Mr. Kirchner,

Thank you for submitting your manuscript to PLOS ONE. After careful consideration, we feel that it has merit but does not fully meet PLOS ONE’s publication criteria as it currently stands. Therefore, we invite you to submit a revised version of the manuscript that addresses the remaining minor points raised during the review process.

We would appreciate receiving your revised manuscript by Jun 06 2020 11:59PM. To enhance the reproducibility of your results, we recommend that if applicable you deposit your laboratory protocols in protocols.io, where a protocol can be assigned its own identifier (DOI) such that it can be cited independently in the future. For instructions see: http://journals.plos.org/plosone/s/submission-guidelines#loc-laboratory-protocols

We look forward to receiving your revised manuscript.

Kind regards,

Hans A Kestler

Academic Editor

PLOS ONE

Reviewers' comments:

Reviewer's Responses to Questions

**Comments to the Author**

1. If the authors have adequately addressed your comments raised in a previous round of review and you feel that this manuscript is now acceptable for publication, you may indicate that here to bypass the “Comments to the Author” section, enter your conflict of interest statement in the “Confidential to Editor” section, and submit your "Accept" recommendation.

Reviewer #1: (No Response)

Reviewer #2: (No Response)

2. Is the manuscript technically sound, and do the data support the conclusions?

Reviewer #1: Yes

Reviewer #2: Yes

3. Has the statistical analysis been performed appropriately and rigorously? 

Reviewer #1: I Don't Know

Reviewer #2: Yes

4. Have the authors made all data underlying the findings in their manuscript fully available?

Reviewer #1: Yes

Reviewer #2: Yes

5. Is the manuscript presented in an intelligible fashion and written in standard English?

Reviewer #1: No

Reviewer #2: No

6. Review Comments to the Author

Reviewer #1: I thank the authors for thoroughly reworking their manuscript in accordance with the suggestions and comments made in my previous review! Only a few issues remain which need to be addressed.

Major comments:

1. Some parts of the discussion (e.g. the reasoning for the guide-RNA concentrations in lines 468-472) should be moved to the Methods section.

2. Sample sizes for the transfection and the Luciferase-Reporter-Assays should be mentioned in the methods section.

3. Is there an analysis on the quality of you laboratory data such as standard deviations of the measurements for the different time points and guide-RNA combinations? If yes, please mention this in the Methods section and the Results section respectively, and also include a table of this data in the manuscript. If no, these analyses should be made and included as described above.

Minor comments:

a) Grammar and punctuation should be revised once more.

b) The whole manuscript should be uniformly written in British OR American English.

c) The formula for the melting temperature of DNA should be mentioned in-line, not as a separate formula in its own line.

Reviewer #2: The paper has been further improved by the second revision. Unfortunately, it is still not appropriate for publication. Therefore, I recommend a minor revision.

First of all, I would like to express my disappointment that the authors are still not able to differentiate between American and British English after repeated hints from both reviewers. The whole manuscript is full of a mixture of both variants. It is particularly noticeable in lines 146-149. It is not so difficult to distinguish between "modelling" (Br.) and "modeling" (Am.), nor between "analyse" (Br.) and "analyze" (Am.). Please revise the whole manuscript especially for these two words. Furthermore, please check the punctuation.

-Line 182-183: Please add a reference for the standard statistical method

-Line 185-186: Please add a reference for AIC

-Remove Fig 7 from the methods. Otherwise you have to change the figure counting.

-Results: Please connect the independently performed experiments with each other. Especially at the beginning line 348-372. It is not clear to the reader why these experiments were done and which conclusions did you draw out of them.

-Line 364-366: No information content. What is the benefit for the reader to know how many data points were collected?

-Line 370-372: Please add here a suggestion/conclusion what does this experiment tells you and do not leave the reader alone with its interpretation.

-Figure description of Fig S1 is missing.

-Take care about the quality of the figures. They appear blurred

7. PLOS authors have the option to publish the peer review history of their article (what does this mean?). If published, this will include your full peer review and any attached files.

Reviewer #1: No

Reviewer #2: No

---

## [Author Response · Author response to Decision Letter 2]

4 Jun 2020

Dear Editor,

thank you for reviewing the revised article and for the opportunity to submit a further revised version of the manuscript. We thoroughly revised the manuscript and worked on all the points raised by the reviewers. We revised the results section in order to describe the results and figures in more detail. Furthermore, the entire manuscript has now been written in British English. 

Yours sincerely

Steven Kirchner

 

Reviewer #1: 

Major comments:

1. Some parts of the discussion (e.g. the reasoning for the guide-RNA concentrations in lines 468-472) should be moved to the Methods section.

Thank you for raising this point. We thoroughly revised the discussion section and moved the paragraph mentioned into the methods section. While carefully revising the manuscript, we did not notice any other sections that had to be moved to the methods section.

2. Sample sizes for the transfection and the Luciferase-Reporter-Assays should be mentioned in the methods section.

Thank you for raising this point. We now mention the sample sizes in the respective paragraph. 

3. Is there an analysis on the quality of your laboratory data such as standard deviations of the measurements for the different time points and guide-RNA combinations? If yes, please mention this in the Methods section and the Results section respectively, and also include a table of this data in the manuscript. If no, these analyses should be made and included as described above.

Thank you for raising this point. We added a table to the supporting information section (S2 Table) that shows REN activation represented by means and standard deviations of all 32 combinations at the three different time points. 

Minor comments:

1. Grammar and punctuation should be revised once more.

We apologize that there were still problems with grammar and punctuation in the second revision of the manuscript. We thoroughly revised the manuscript in order address those issues. 

2. The whole manuscript should be uniformly written in British OR American English.

We apologize that there were still problems with the language in the second revision of the manuscript. The whole manuscript should now be written in British English.

3. The formula for the melting temperature of DNA should be mentioned in-line, not as a separate formula in its own line.

Thank you for raising this point. The formula for the melting temperature of DNA is now mentioned in-line. 

 

Reviewer #2: 

1. Please revise the whole manuscript especially for these two words (“modelling”, “analyse”). Furthermore, please check the punctuation.

We apologize that there were still problems with language, grammar and punctuation in the second revision of the manuscript. We thoroughly revised the manuscript in order address those issues. The whole manuscript should now be written in British English.

2. Line 182-183: Please add a reference for the standard statistical method

Thank you for raising this point. We added the following reference: 

Fisher RA. The Goodness of Fit of Regression Formulae, and the Distribution of Regression Coefficients. Journal of the Royal Statistical Society. 1922;85(4):597-612

DOI: 10.2307/2341124

3. Line 185-186: Please add a reference for AIC

Thank you for raising this point. We added the following reference: 

Akaike H. CITATION CLASSIC - A NEW LOOK AT THE STATISTICAL-MODEL IDENTIFICATION. Current Contents/Engineering Technology & Applied Sciences. 1981(51):22-

DOI: 10.1109/TAC.1974.1100705

4. Remove Fig 7 from the methods. Otherwise you have to change the figure counting.

Thank you for raising this point. We removed “The distribution of the errors of the individual experiments is shown in Fig 7” from the methods section. 

5. Results: Please connect the independently performed experiments with each other. Especially at the beginning line 348-372. It is not clear to the reader why these experiments were done and which conclusions did you draw out of them.

Thank you for raising this point. We thoroughly revised the results section in order to address this issue and to give the reader more details to understand the performed experiments. 

6. Line 364-366: No information content. What is the benefit for the reader to know how many data points were collected?

Thank you for raising this point. We removed the respective sentence from the manuscript. 

7. Line 370-372: Please add here a suggestion/conclusion what does this experiment tells you and do not leave the reader alone with its interpretation.

Thank you for raising this point. We added the following paragraph to the manuscript:

“… In Fig 3 expression levels of the 32 possible combinations are shown at three different time points after transfection using the modified Venn diagram as described above (Fig 2). Respective activation levels of REN were expressed in a colour ramp that rose in ascending order from white to yellow, further to red and up to blue. For analysis, the values of expression levels were scaled logarithmically. Otherwise, since these values were widely divergent, only a few surfaces would have been coloured in a linear scale, whereas most were white to slightly yellow. On the one hand the individual guides itself caused different levels of renin expression. On the other hand, the different combinations achieved different renin activation (Fig 3 and S2 table). The combinations of the two guides C – B or A – E caused a renin activation that was higher than the simple summation of the respective individual activation levels. In contrast, combinations of the two guides A – B or also D – C have a less increasing effect on the renin activation. …”

8. Figure description of Fig S1 is missing.

Thank you for raising this point. We added the following description for Fig S1:

“The cassette contains the Firefly luciferase, which was used as a reporter for REN expression. Puromycin was used to select the cells. The homology arms were required for the correct in frame insertion of the cassette. The elements G418-resistance and thymidine-kinase also contained in the cassette were not used in this study.”

9. Take care about the quality of the figures. They appear blurred

Thank you for raising this point. We followed the instructions of PLOS ONE to upload the figures. It might be that the conversion into pdf reduces the quality of the figures. For a potential production we would have all the pictures available in high quality.

---

## [Decision Letter · Decision Letter 3]

18 Jun 2020

Decipher the complexity of cis-regulatory regions by a modified Cas9

PONE-D-19-26748R3

Dear Dr. Kirchner,

We’re pleased to inform you that your manuscript has been judged scientifically suitable for publication and will be formally accepted for publication once it meets all outstanding technical requirements.

Kind regards,

Hans A Kestler

Academic Editor

PLOS ONE

Additional Editor Comments (optional):

Reviewers' comments:

Reviewer's Responses to Questions

**Comments to the Author**

1. If the authors have adequately addressed your comments raised in a previous round of review and you feel that this manuscript is now acceptable for publication, you may indicate that here to bypass the “Comments to the Author” section, enter your conflict of interest statement in the “Confidential to Editor” section, and submit your "Accept" recommendation.

Reviewer #1: All comments have been addressed

Reviewer #2: All comments have been addressed

2. Is the manuscript technically sound, and do the data support the conclusions?

Reviewer #1: Yes

Reviewer #2: Yes

3. Has the statistical analysis been performed appropriately and rigorously? 

Reviewer #1: Yes

Reviewer #2: Yes

4. Have the authors made all data underlying the findings in their manuscript fully available?

Reviewer #1: Yes

Reviewer #2: Yes

5. Is the manuscript presented in an intelligible fashion and written in standard English?

Reviewer #1: Yes

Reviewer #2: Yes

6. Review Comments to the Author

Reviewer #1: I thank the Authors for their continued work on the manuscript. All of my previous concerns have been addressed.

I have one last minor comment to make:

In Figure 4 "cps" is mentioned but it is never explained what this abbreviation means. Please include this information in the Figure text, and please check the rest of the manuscript for more unexplained abbreviations (I might have missed one in my read-through).

Reviewer #2: The manuscript has improved a lot and is now suitable for publication.

Please correct the below mentioned spelling mistakes:

-line255: add punctuation

-line 269: use "," to separate 1000  50,000

-line 364: VisualiSation

7. PLOS authors have the option to publish the peer review history of their article (what does this mean?). If published, this will include your full peer review and any attached files.

Reviewer #1: No

Reviewer #2: No

---

## [Editor Report · Acceptance letter]

24 Jun 2020

PONE-D-19-26748R3 

 Decipher the complexity of cis-regulatory regions by a modified Cas9 

Dear Dr. Kirchner:

I'm pleased to inform you that your manuscript has been deemed suitable for publication in PLOS ONE. Congratulations! Your manuscript is now with our production department. 

Kind regards, 

on behalf of

Prof. Hans A Kestler 

Academic Editor

PLOS ONE